# A new multicopter based unmanned aerial system for pollen and spores collection in the atmospheric boundary layer

Claudio Crazzolara[1,2], Martin Ebner[1], Andreas Platis[1,2], Tatiana Miranda[1,3], Jens Bange[1,2], Annett Junginger[1,3]

[1]Department for Geosciences, Eberhard-Karls-Universität Tübingen, 72074 Tübingen, Germany
[2]Center for Applied Geoscience, Eberhard-Karls-Universität Tübingen, 72074 Tübingen, Germany
[3]Senckenberg Centre for Human Evolution and Paleoenvironment (S-HEP-Tübingen), 72074 Tübingen, Germany

*Correspondence to*: Claudio Crazzolara (claudio.crazzolara@uni-tuebingen.de)

## Abstract

The application of a newly in-house developed particle collection system (PCS) onboard a commercially
available multicopter unmanned aerial vehicle (UAV) is presented as a new unmanned aerial system (UAS) approach for in-situ measurement of the concentration of aerosol particles such as pollen grains and spores in the atmospheric boundary layer (ABL). A newly developed impactor is used for high efficiency particle extraction onboard the multicopter UAV. An air volume flow of 0.2 m$^3$ per minute through the impactor is provided by a battery powered blower and measured with an onboard mass flow sensor. A bell mouth shaped
air intake of the PCS is arranged and oriented on the multicopter UAV to provide substantially isokinetic sampling conditions by advantageously using the airflow pattern generated by the propellers of the multicopter UAV.

More than thirty aerosol particle collection flights were carried out near Tübingen in March 2017 at altitudes of up to 300 m above ground level (a.g.l.), each with a sampled air volume of 2 m$^3$. Pollen grains and spores
of various genera as well as large (>20 µm) opaque particles and fine dust particles were collected and specific concentrations of up to 100 particles per m$^3$ were determined by visual microscopic analysis. The pollen concentration values measured with the new UAS match well with the pollen concentration data published by the Stiftung Deutscher Polleninformationsdienst (PID) and by MeteoSchweiz. A major advantage of the new multicopter based UAS is the possibility of the identification of collected aerosol
particles and the measurement of their concentration with high temporal and spatial resolution, which can be used inter alia to improve the data base for modelling the propagation of aerosol particles in the ABL.

# 1 Introduction

In-situ measurements of the concentration of aerosol particles such as pollen, spores, and fine particulate matter in the atmospheric boundary layer (ABL) are of great interest in numerous scientific disciplines (Hardin & Hardin, 2010).

For example, in agricultural science, the concentration and aerial dispersal of pollen and spores is of interest with regard to an optimization of yield (Aylor, 2005), the spread of plant diseases (Aylor et al., 2011), and also with regard to the spread of transgenic material originated from genetically manipulated corn (Hofmann et al., 2013). In particular, plant pathogens are able to travel hundreds or even thousands of kilometres through the atmosphere from their origin to the place where they cause damage (Schmale & Ross, 2015). The travel distance but also concentration of pollen is furthermore dependent on the seasonal atmospheric convective conditions (Boehm et al., 2008). For example, seasonal variations have been reported for fungal spores of the genus *Fusarium* (Lin et al., 2014) with distributions in altitudes of 40 to 320 metres above ground level (a.g.l.) as reported by Schmale et al. (2012) using an unmanned aerial vehicle.

In meteorology, it is known that mineral dust particles originated from Saharan dust storms and transported for example to Southern Florida effectively act as ice nuclei being capable for glaciating super cooled altocumulus clouds (Sassen et al., 2003). Pollen grains, although being only moderately hygroscopic, are able to act as cloud condensation nuclei and exhibiting a bulk uptake of water under subsaturated conditions (Pope, 2010). Investigations on the hygroscopic growth of pollen suggest that extreme pollen concentrations ($> 1,000$ m$^{-3}$) may interfere with the activation of the background sulphate aerosol mode in pristine environments (Griffiths et al., 2012). Also spores of which millions of tons are dispersed into the atmosphere every year, may act as nuclei for condensation of water in clouds (Hassett et al., 2015). It is also suggested, that some atmospheric microbes could catalyse the freezing of water at higher temperatures and may facilitate the onset of precipitation (Jimenez-Sanchez et al., 2018). Thus, the knowledge about the spatial distribution and transportation distances of dust particles, pollen, spores, and microbes would allow the determination of their contribution in cloud formation processes, which are influencing not only local weather, but also regional or even worldwide climate. Meteorological processes have a great influence on the propagation behaviour of the aerosol particles in the ABL. For in situ measurements of relevant meteorological parameters in the ABL, e.g. the air temperature with high temporal resolution, a remotely piloted fixed-wing unmanned aerial vehicle (UAV) can be used (Wildmann et al., 2013). Also, the use of a multicopter UAV with onboard temperature, humidity and gas sensors for in situ measurements of meteorological variables in the ABL was reported recently (Brosy et al., 2017).

In human medicine, the careful scientific evaluation of the actual concentration of pollen in the air is the indispensable basis for reliable pollen risk information. Inadequate forecasts concerning the expected pollen concentration are regarded as a considerable health risk for pollen allergy sufferers (Bastl et al., 2017). Damialis et al. (2017) reported just recently of the first basic experiments measuring pollen concentrations in

considerable altitudes above ground level by using a manned aircraft. However, this research has shown, that the use of manned aircraft in densely populated areas is limited and further requires a considerable organizational and financial effort.

In environmental sciences, the pollution of air with fine particulate matter has been a problem for many years, in particular in urban areas with unfavourable geographical topography. The so-called PM2.5 and PM10 particulate matter according to the National Air Quality-Standard for particulate matter of the U.S. Environmental Protection Agency (Vincent, 2007) as well as coarse particles have been chemically characterized by Hueglin et al., 2005. In simplified view, PM2.5 is the fraction of particulate matter (PM) consisting of inhalable particles having a size of 2.5 µm and smaller, whereas PM10 is the fraction of particulate matter (PM) consisting of inhalable particles having a size of 10 µm and smaller. Accordingly, PM2.5 is incorporated in PM10. The samples were taken using pre-weighed quartz fibre filters, which were weighed again after collection of particles. This method requires considerable expenditure and processing time in particular for pre- and re-conditioning of the filters prior to the respective weighing step. The possibility of assigning health risks to specific classes of particulate matter has been investigated, but the results are not satisfactorily reliable yet (Harrison and Yin, 2000), not least because of the scarcity of measurement data, which are, in turn, related to the complex measuring methods. Further areas of greater interest in particle concentration in the air are the scientific fields of paleo-environmental and paleo-climatological reconstructions. Here, for example, the knowledge of the spatial and temporal distribution of pollen could help to gain insights in their genus-specific propagation behaviour and possible transport distances. This would enable to improve the accuracy of paleoclimate models derived from pollen grains extracted from lacustrine or marine sediments (Shang et al., 2009).

For most of these applications, it would be highly desirable not only to count the number or measure the size of the particles as done with an optical particle counter (OPC), but also to identify the particles according to their type and/or chemical composition. In this regard, particle collection with subsequent particle-type identification and quantification is of advantage over particle counting at least as long as reliable in-situ particle identification is not available. First attempts to collect bioaerosol particles using a pollen trap mounted on a fixed wing UAV are described in Gottwald and Tedders (1985). Another way to realize the collection of airborne particles is to use a tethered balloon with rotating rods for capturing airborne pollen grains (Comtois et al., 2000). Since the balloon experiences wind drift, the possibilities of performing measurements at a predetermined position are limited. In addition, the air volume sampled by the rotating rods is determinable with limited accuracy only. Sticky surfaces carried by a fixed-wing autonomous UAV described by Schmale et al. (2008) and Aylor et al. (2011) allow long-range particle collection but provide only limited spatial resolution of particle concentration values. The sampled air volume, again, is determinable with limited accuracy only. In addition, the requirement of a runway for start and landing limits the potential use of fixed wing UAVs in urban or built-up areas.

Here we present the structural design and first application of a newly in-house developed particle collection

system (PCS) operated onboard a commercially available multicopter UAV (Fig. 1) for in situ measurements of the concentration of pollen and spores in the ABL. Initially, a commercially available multicopter UAV that meets the requirements for payload capability as well as flight stability and reliability was selected and built from a kit. The multicopter UAV provides not only the possibility of the vertical take-off and landing, thus simplifying the application in urban areas, but – even more important – also the possibility of hovering and hence collecting particles at elevated positions that can be maintained with high precision. Then experiments were conducted to investigate the air flow pattern created by the UAV's propellers during hovering. The experimental results were used to determine the dimension and position of the air intake of the PCS on the multicopter UAV in order to provide substantially isokinetic sampling conditions.

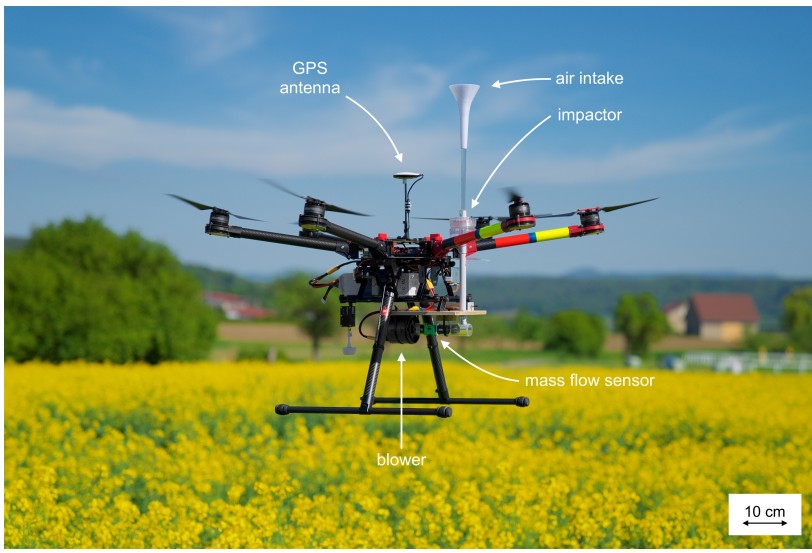

**Figure 1.** Multicopter UAV (DJI S900) in hovering flight with components of the particle collection system as indicated: air intake, impactor, mass flow sensor, and blower. The inlet of the air intake is arranged about 30 cm above the propeller plane.

An essential part of the present study was the development of a new PCS that can be operated onboard the multicopter UAV despite the weight and power constraints. One major goal in the development of the PCS was to sample an air volume of 1 $m^3$ within 5 minutes in order to ensure a statistically evaluable number of collected particles even in the case of low particle concentrations in the air, and also to provide a high temporal resolution of the measurement results compared to other particle collection systems. This goal was achieved by using a powerful blower that delivers an air volume flow of typically 0.2 $m^3$ per minute (corresponding to 200 thousand standard cubic centimetres per minute – 200,000 sccm) through the PCS. Another challenge was to develop an impactor that ensures reliable separation of the aerosol particles even at these high air flow rates.

In order to determine the capability of the PCS operated onboard the multicopter UAV and to test the reliability of the entire new unmanned aerial system (UAS), several test flights were conducted at different altitudes over several days in March 2017. The collected particles were analysed and counted using light microscopy. Finally, the pollen concentration values determined with the PCS onboard the multicopter UAV were compared with corresponding data published by forecast information services such as the Stiftung Deutscher Polleninformationsdienst (PID) or MeteoSwiss.

## 2 Development of a system for aerial particle collection

### 2.1 Multicopter unmanned aerial vehicle (UAV)

A DJI S900 hexacopter, commercially available from the Chinese company DJI Technology Co. Ltd, was selected as multicopter UAV with regard to flight performance, payload capabilities, and expansion options. The DJI S900 has a diagonal wheelbase of 900 mm and a maximum take-off weight of 8.2 kg. Propeller arms and propellers are foldable, which allows a space saving and comfortable transport and fast set up time of less than 10 minutes at the site of operation including the set-up of the PCS. At ambient air temperatures between -5 °C and +37 °C as experienced during tens of flight operations in 2017, the DJI S900 worked reliably, i.e. not a single flight interruption due to technical problems occurred, and it was robust, i.e. the components withstood all applied stresses without any problems or hardware failure.

A DJI A2 flight control system was employed to automatically control the flight attitude, i.e. roll, pitch, and yaw angles as well as the flight altitude, and to maintain the spatial position of the multicopter UAV using a GPS receiver. A remote control of the type T14SG (2.4 GHz band, 14 control channels) by Futaba Corporation was chosen due to its high reliability over long distances. Telemetry data such as battery parameters (voltage, current, and capacity) and the barometrically determined flight altitude above ground level were re-transmitted from the remote-control receiver onboard the multicopter UAV to the handheld transmitter on the ground.

The DJI S900 was operated with a 6-cell Lithium polymer battery (LiPo, 22.2 V, 12,000 mAh, 266 Wh). During regular flight operations, preferably only 80 % of the nominal capacity was taken from the battery in order to have safety reserves in case of unexpected flight manoeuvres and to increase the durability of the LiPo-battery. The fully equipped multicopter UAV including the mounted PCS has a take-off weight of 6.5 kg. The possible flight time is dependent on several factors including the altitude above sea level (a.s.l.) of the launch site, the prevailing wind conditions, and the altitude above ground level (a.g.l.) during particle collection operation. For our aerosol particle collection flights, starting from a launch site 400 m a.s.l. with side winds on the ground of about 2 m/s, typical flight times were 15 minutes including a 10-minute aerosol particle collection operation at an altitude of 300 m a.g.l., while the remaining battery capacity was typically 30 %.

## 2.2 The set-up of the new particle collection system (PCS)

A new PCS was developed in order to meet the requirements for aerial use onboard a multicopter UAV. To ensure a number of at least 10 collected particles even in the case of a particle concentration in the sampled air being as low as 5 particles per $m^3$, an air volume of 2 $m^3$ has to be sampled. With regard to the limited maximum flight time of the multicopter UAV, typically 10 minutes are available for airborne particle collection operation. Accordingly, the PCS has to be able to process an air volume flow of 0.2 $m^3$ $min^{-1}$.

Starting from these boundary conditions, an impactor-based PCS was developed (Fig. 2) that comprises: (1) an air inlet that allows the intake of ambient air under near-isokinetic conditions, (2) an impactor for extracting the particles from sampled air and depositing them on a sample carrier, (3) a mass flow sensor, located downstream of the particle extractor, measuring the air mass flow through the PCS, and (4) an electric blower generating the air flow through the components of the PCS independent of the airspeed of the multicopter UAV. The components of the PCS and their connections are leak-tight, which means that the air volume passing the mass flow sensor is the same that is flowing through the particle extractor and the same as the air volume taken in at the air intake.

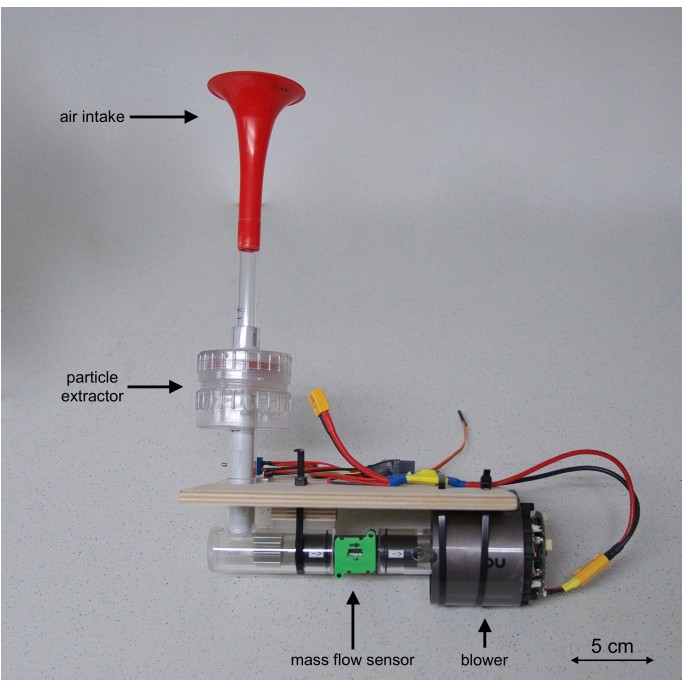

**Figure 2.** Newly developed particle collection system (PCS) with a complete weight of 600 g comprising (1) an air intake that allows the intake of ambient air under near-isokinetic conditions, (2) an impactor for extracting the particles from sampled air and depositing them on a sample carrier, (3) a mass flow sensor, located downstream of the particle extractor, measuring the air mass flow through the PCS, and (4) an electric blower generating the air flow through the components of the PCS. The components of the PCS and their connections are leak-tight. Air volume flow during operation is 200 slm.

### 2.2.1 Bell mouth shaped air intake

The geometry and orientation of the air intake must be chosen in such a way that the sampled air is representative in terms of its particle load, which can be achieved by so-called "isokinetic" sampling (Kulkarni et al., 2011). Isokinetic sampling means that the flow velocity of the air entering the air intake is identical, by magnitude and direction, to the flow velocity of the ambient air approaching the air intake. If an isokinetic sampling is not assured, effects based on the aerodynamic behaviour of aerosol particles, such as their mass inertia and coefficient of drag $c_d$, can result in a particle uptake of the ambient air that is not representative and leads to a falsification of the measured particle concentration value. The larger the particles are and the more mass and thus inertia they have, the more important isokinetic sampling becomes (Kulkarni et al., 2011).

In order to provide an omnidirectional air intake under isokinetic or at least near-isokinetic conditions, a bell mouth was chosen as the shape of the air intake with a wide end for the air inlet and a narrow end for the connection to the subsequent particle extraction unit (Fig. 2). The substantially hyperbolic form continuously accelerates the air that is drawn in. While the velocity of the air entering the air intake at the wide end is typically 1 to 3 m s$^{-1}$, the air is accelerated to a mean velocity of 50 m s$^{-1}$ at the narrow end.

### 2.2.2 Impactor as particle extraction unit

Operation on a multicopter UAV requires a particle extraction unit that has a low mass and provides high particle extraction rate even at large air volume flows (0.2 m$^3$ min$^{-1}$) in order to allow short (10 min) sampling operation periods. Additionally, in order to achieve a lean workflow from sampling to visual particle identification and counting, the extracted particles should be easily accessible for visual analysis without complex and time-consuming sample preparation steps. In this context "lean workflow" also means that preferably an initial estimate of the quantity and type of particles collected should be possible already in the field by visual inspection with simple tools such as a magnifying glass; this allows, if necessary, an adjustment of the flight altitude or the sampling operation period during the immediately following particle collection flight. A device that has the potential to meet all these demands is based on a so-called impactor.

The functional principle of an impactor is based on the deflection of a particle-loaded free-flow gas stream by means of an impaction plate (Kulkarni et al., 2011). The gas stream is usually accelerated through a nozzle up to a velocity that is depending on the volume flow and nozzle geometry. An impaction plate coated with an adhesive film is arranged in the open jet at a small distance from the nozzle that forces the particle-loaded gas stream to deflect. Due to their mass inertia, the particles in the gas stream are able to follow this deflection only to a limited extent. As a consequence, particles with a sufficiently high mass inertia impinge on the surface of the impaction plate and are retained on the adhesive film. Hirst (1952) first described the application of an impactor-based device for extracting aerosol particles such as spores, but only for stationary use and sampling of a very low air volume flow of about 10 litres per minute.

In order to sample an air volume of 2 m$^3$ within an aerial sampling operation period of 10 minutes, a sampled air volume flow of 0.2 m$^3$ (200 litres) per minute is required. The orifice of the impactor was chosen to be circular shaped with a diameter of 9 mm, corresponding to an orifice area of about 64 mm$^2$. Thus, for an air volume flow of 0.2 m$^3$ per minute, the mean velocity of the open jet in the orifice area is about 50 m s$^{-1}$. This

mean velocity $v_a$ through the orifice area $A$ can be calculated from the volume flow $Q$ and the area $A$ by $v_a = Q/A$.

Figure 3 shows a longitudinal cut through the newly developed impactor of the PCS. A commercially available 50 mm diameter filter housing from the Sartorius AG was used with modifications to form the case of the impactor. The housing comprises two injection-moulded halves of transparent polycarbonate (PC)

forming an upper and a lower part that can be screwed together. Into a central bore of the upper part of the filter housing, the lower end of a first transparent polymethyl methacrylate (PMMA) cylindrical pipe with an inner diameter of 9 mm was inserted. The upper end of the first pipe will be connected to the bell mouth shaped air intake. Into a central bore of the lower part of the filter housing, the upper end of a second PMMA cylindrical pipe with an inner diameter of 16 mm was inserted; the lower end of the second pipe will be

connected to the mass flow sensor as described in the following section. In between the two housing halves, a particle sample carrier acting as the impaction plate was installed opposite the lower end of the first cylindrical pipe.

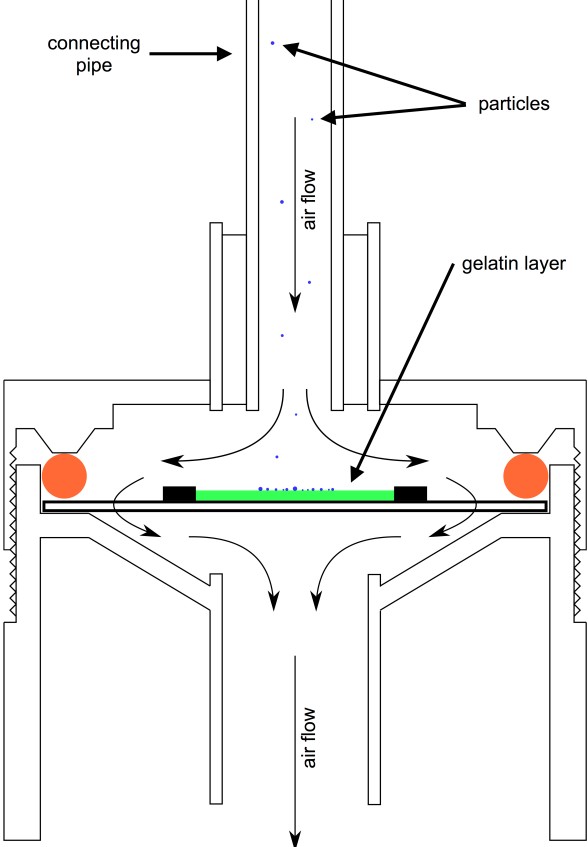

**Figure 3.** Schematic longitudinal cross section through the impactor used as a particle extractor in the particle collection system. Particles are drawn through the pipe from the top towards the glycerine gelatine covered microscope slide. Glycerine gelatine highlighted in green, cross section of silicone O-ring in red. Mean impaction velocity is about 50 m s$^{-1}$.

5 The particle sample carrier is 43.5 x 26 mm in size and 1 mm thick and can be cut from a conventional microscopic glass slide. An adhesive film of glycerine gelatine was applied onto the glass slide in order to retain the impinged particles. Details on slide preparation are described in 2.3. The sample carrier rests in the lower housing part on a circular ring-shaped surface (Fig. 3). When the two housing parts are screwed together, the particle sample carrier is fixed by means of a silicone O-ring, which rests on the sample carrier

10 and is pressed down by the upper housing part as shown in Figure 3. Figure 4A shows a perspective view on the assembled particle extractor, while Figure 4B shows a perspective view on the particle extractor with the upper housing part removed, and Figure 4C shows a top view on the particle extractor with the upper housing part removed and with a particle loaded sample carrier. The total weight of the impactor including the upper and lower pipes and the installed particle sample carrier is about 50 g.

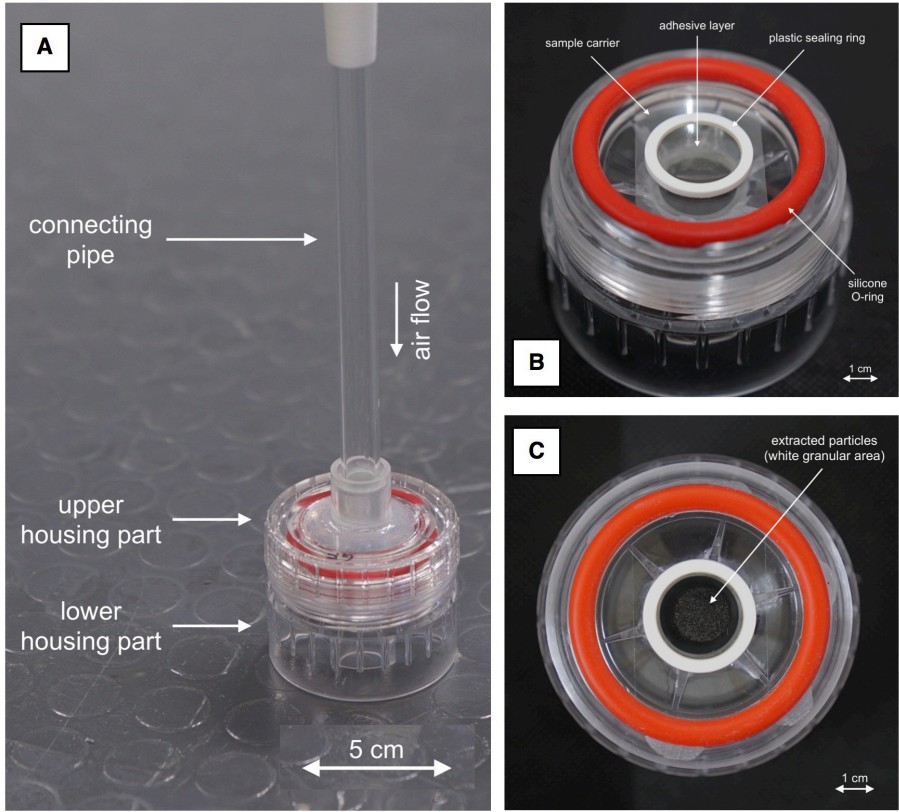

**Figure 4.** (A) Perspective view on the assembled particle extractor, with connecting pipe being connected into the bellmouth-shaped air intake; (B) perspective view on the particle extractor with the upper housing part removed, and the sample carrier installed; (C) top view on the particle extractor with the upper housing part removed and particle loaded sample carrier; extracted particles are deposited in the area enclosed by the white plastic ring.

### 2.2.3 Mass flow sensor

A reliable determination of the concentration of aerosol particles requires the precise determination of the sampled air volume. This was achieved by installing a mass flow sensor that permanently remains in the air flow path of the PCS, irrespective of whether data from the flow sensor were collected or not. A SFM 3000-200-C mass flow sensor of Swiss company Sensirion AG was used for this purpose. This sensor offers a bi-directional measuring span of +/- 200 standard litres per minute (slm), with standard conditions defined as 20 °C air temperature and 1,013.25 hPa, and provides a digital output signal using the $I^2C$ protocol. The accuracy of the individually calibrated sensor is 1.5 % (typical) and 2.5 % (maximum) of measured value between -20 °C and +80 °C, and the update time is 0.5 ms corresponding to 2,000 Hz. The total weight is 18 g with the dimensions of 100 mm x 20 mm x 30 mm (length x width x height).

### 2.2.4 Blower

The electrically operated blower must ensure high air volume flow through the PCS during flight operations and the associated power and mass limitations. It is also necessary that the blower performance is substantially independent of fluctuations of the battery voltage in order to provide a constant air volume flow through the PCS. A blower that meets these demands is commercially available in handheld vacuum cleaners of British company Dyson Limited. The blower that we used in the PCS has a total weight of 245 g and can be operated in two power levels, either 100 or 350 W. Due to its integrated microprocessor control, the blower features a very fast spin up (0.2 s) and spool down time (1 s), and provides constant blower power in a battery voltage range between 20.4 and 25.2 V. An adjustable leak valve is arranged in the connection between mass flow sensor and blower since the blower offers a considerable surplus already if operated in the lower 100 W mode. On ground, the leak valve was adjusted to set the air volume flow to 200 slm by digitally reading out the mass flow sensor. As regularly performed control measurements have shown, this setting is very stable over many measurement flights. One channel of the remote control system was used to switch the blower on and off when the multicopter UAV was airborne and the particle collection position, i.e. the desired altitude a.g.l., was reached. As long as the blower was switched on, i.e. as long as particle collection was performed, the multicopter UAV was maintained (by hovering) in this desired particle collection position. Before leaving this position, the blower was remotely switched off, thus terminating the particle collection operation. The value of the electrical current, drawn by the blower from the battery, was measured onboard the multicopter UAV and transmitted to and monitored on ground, to make sure that the blower has really went into operation (switched on) or was really out of operation (switched off).

### 2.3 Preparation and handling of sample carriers

An individual particle sample carrier was used for each particle collection operation (Fig. 5A). Accordingly, after each particle collection operation, the sample carrier was removed from the impactor and replaced by a new one. The particle sample carrier consists of a common microscope glass slide with a size of 43.5 mm by 26 mm. An adhesive layer of glycerine gelatine (Morphisto Evolutionsforschung und Anwendung GmbH, Frankfurt, Germany) was applied circularly on the surface of the glass plate facing towards the open jet allowing the aerosol particles to penetrate onto the sticky surface. In order to define and limit the lateral extent of the gelatine layer, a circular sealing ring made of polyamide (PA) with an inner/outer diameter of 17/22 mm, a thickness of 1.5 mm and a rectangular cross section was arranged centrally on the glass plate. The glycerine gelatine was heated in a water bath at 45 °C and poured onto the glass plate into the circular area delimited by the polyamide sealing ring.

The sample carriers were produced in batches, usually a few days prior to the scheduled particle sampling operation with the production date of the batches being recorded. Production, handling, and storage of the sample carriers were performed in a portable laminar air flow box under continuous flow of filtered air. The air was filtered by two pre-filters and finally a H14-specified HEPA (High Efficiency Particulate Air) filter

removing more than 99.995 % of the particles in the most critical size range of 0.1 to 0.3 µm. Small containers of transparent plastic were used for individually transporting and storing the particle sample carriers prior and post particle sampling operation. Repeated inspections proved that these measures reliably prevent contamination of sample carriers during manufacture, handling, transport, and storage.

Careful post-sampling treatment is highly necessary to avoid contamination and allow preservation. Immediately after landing the multicopter UAV, the particle-loaded sample carrier was carefully removed from the impactor and placed into its transport box (Fig. 5B, step 1). Back in the laminar air flow box in the lab, a protective layer of one drop liquid gelatine was applied onto the particle-loaded gelatine layer (Fig. 5B, step 2) in order to prevent damage to the particle-loaded gelatine layer. A common microscope cover slip

(22 x 22 mm, 0.15 mm thick) was then placed centrally on the liquid gelatine in order to protect and seal the sample from contamination (Fig. 5B, step 3). Finally, this cover slip was lowered gently vertically allowing the liquid gelatine to spread (Fig. 5B, step 4). Special care was taken to avoid air bubbles between the cover slip and the gelatine.

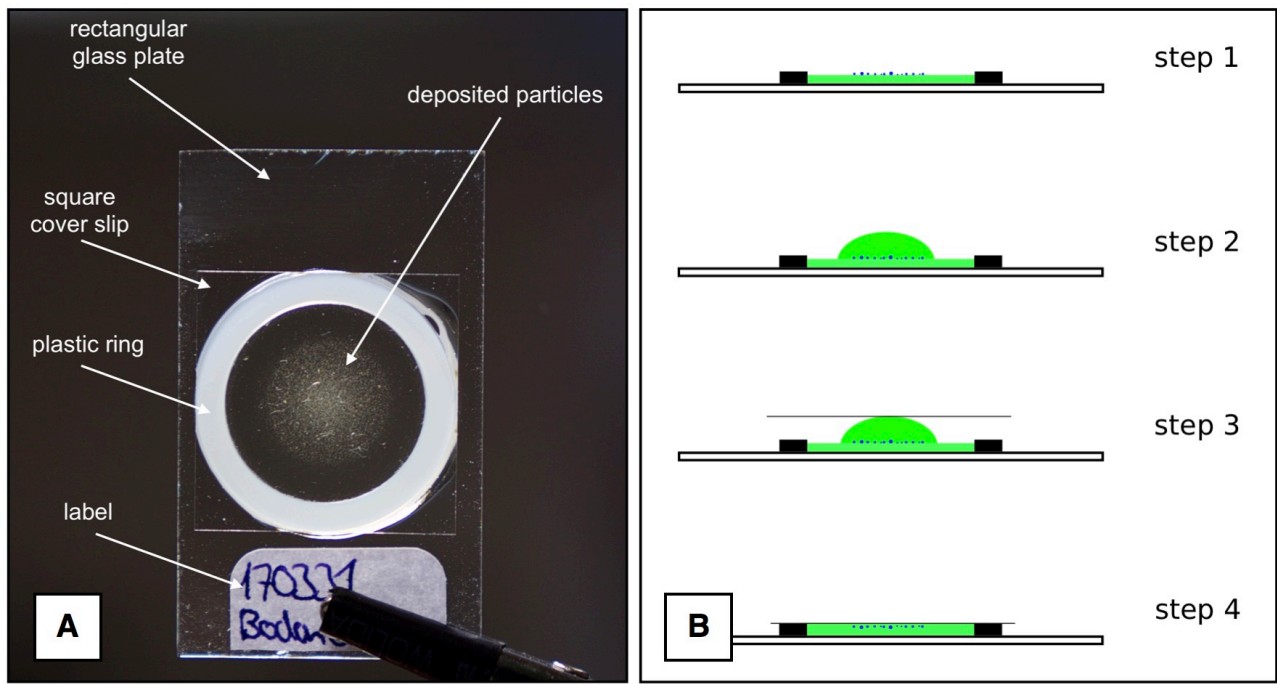

**Figure 5.** (A) Top view on a particle loaded sample carrier comprising a common microscope slide and a plastic ring with gelatine used as the particle embedding layer, covered with a transparent microscope cover slip (square). (B) Post-sampling treatment steps 1 to 4 of the particle loaded sample carrier to avoid contamination and allow preservation, shown as cross sections through a sample carrier. Highlighted in green: gelatine layer in which the collected particles (blue dots) are embedded. Step 1: Sample carrier immediately after particle collection with deposited particles exposed; Step 2: A drop of molten gelatine is placed onto the particle-

loaded gelatine layer; Step 3: A cover slip is placed centrally on the drop of liquid gelatine; Step 4: The cover slip is lowered vertically to protect and seal the particle-loaded glycerine gelatine.

# 3 Experiments

## 3.1 Multicopter caused air flow pattern (Smoke Plume Test)

When using a multicopter UAV for aerosol particle collection, it needs to be considered where the air intake of the PCS has to be positioned. It also needs to be considered how the air intake should be aligned in relation to the airflow generated by the propellers of the multicopter UAV in order to avoid an impairment of the measurement results and to ensure a substantially isokinetic sampling. Haas et al. (2014) used computational fluid dynamics (CFD) calculations for a complete study of the aerodynamics of a multicopter UAV of a similar size and weight to the one used in the presented study. As a result of their CFD-calculations, the volume of air mixed by the propellers of the multicopter UAV is approximately a cylinder with a radius of 2 m and with an extent of 2 m above and 8 m below the multicopter UAV. Calculations of the magnitude of air velocity showed high values in the immediate vicinity of the propellers as well as below the propellers, whereas the corresponding values above the propellers are significantly lower. Thus, for the collection of aerosol particles as intended within this study, it was decided to arrange the air intake of the PCS sufficiently above the propellers of the multicopter UAV.

In order to investigate the actual airflow around the multicopter UAV used in this study under ambient conditions with side wind, a visual air flow test was performed in January 2017 at the airfield in Poltringen, Germany (48.54322° N, 8.94865° E, 400 m a.s.l.). For this purpose, three coloured pyrotechnical smoke cartridges (type AX 60, company BJÖRNAX AB, Nora, Sweden) were mounted and ignited at different positions on an erectable aluminium boom with the multicopter UAV flying at different elevations below and above the generated smoke plumes (Fig. 6). The whole experiment was filmed and the video sequences were analysed with regard to the resulting air flows.

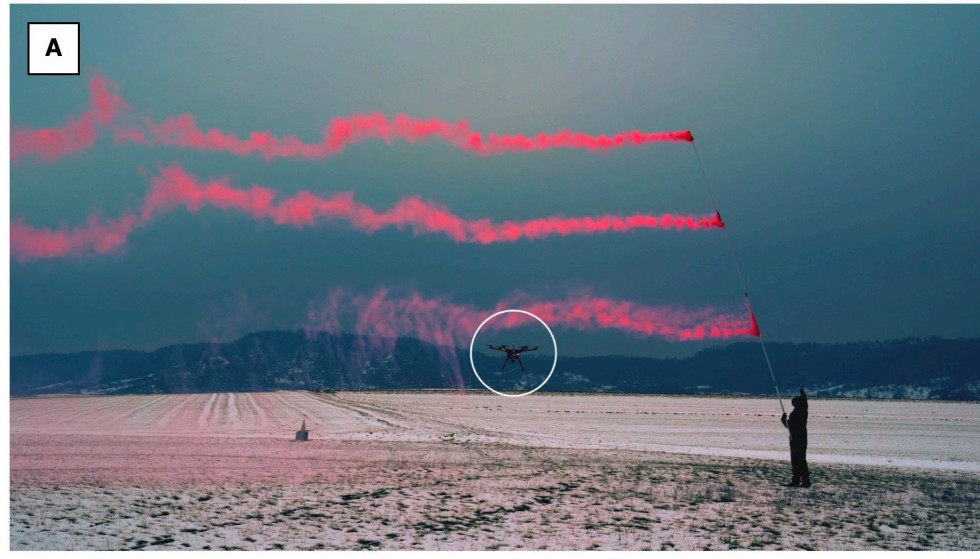

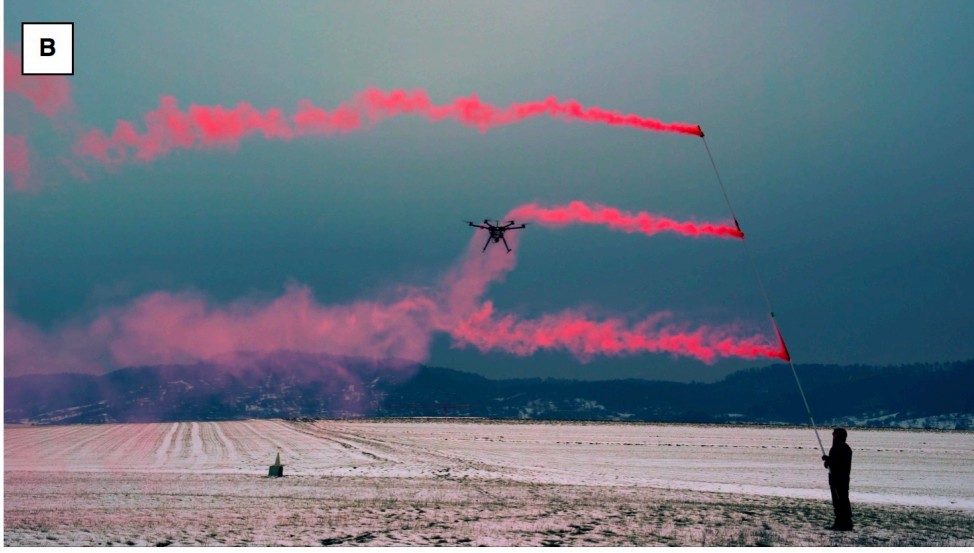

**Figure 6.** Investigation of the air flow pattern caused by the multicopter UAV (DJI S900) using three coloured pyrotechnical smoke cartridges with (A) flying the multicopter UAV below the lowest smoke plume, and (B) below the middle smoke plume; screen shots taken out of a 30 seconds video sequence. Side wind from right to left. Dilution of the smoke plume and thus mixing of the surrounding air occurs essentially only on the lee side and below the multicopter UAV, while in windward and above the multicopter UAV, the approaching plume remains largely unaffected.

Figure 6A shows that only the first (lowest) smoke plume approaching (due to prevailing side wind) horizontally about 80 cm above the multicopter UAV is influenced by the downwash caused by the propellers and accelerated vertically downwards. The second smoke plume (2.4 m above the multicopter UAV) and the third smoke plume (4.0 m above the multicopter UAV) remain substantially unaffected.

Furthermore, it is also shown that the first smoke plume is greatly diluted on the lee side (with respect of the side wind blowing from right to left) of the multicopter UAV, which is a result of the downward acceleration of the associated air mass. The upper second and third smoke plumes also experience some turbulence on the lee side but significantly less than the first smoke plume. As a result, the air mass on the lee side of the multicopter UAV seems to be much more effected by the downwash caused by the propellers than the air mass windward.

Figure 6B shows a photograph with the multicopter UAV elevated only about 20 cm below the second smoke plume. It can be seen that the second smoke plume is directly captured by the propellers of the multicopter UAV. Thus, the second smoke plume is accelerated and accordingly diluted downwards. Also, the lower first smoke plume is heavily affected and disturbed by the downwash caused by the propellers of the multicopter UAV, whereas the upper third smoke plume (1.8 m above the multicopter UAV) remains almost unaffected. For the present study, the dilution of the smoke plume was not of interest per se. Instead, the velocity (by magnitude and direction) of characteristic patterns of the smoke plume approaching the multicopter UAV was of interest, in order to decide where the air intake of the PCS has to be arranged and how it has to be oriented to achieve substantially isokinetic sampling conditions. The results are discussed in 4.1.

### 3.2 Particle extraction efficiency of the impactor (Cascade Test)

In order to examine the effectiveness of the newly developed PCS with respect to its particle extraction rate, an experiment was carried out using two identical impactors connected in a cascade (Fig. 7). The experiment was carried out on ground with the same operating conditions as during particle collection flights in order to ensure the comparability of the results. Prior to this experiment, all impactor housing and tubing components were carefully cleaned to ensure that all components used in these experiments are particle-free. Fresh sample carriers were inserted in both impactors. Then the blower was operated for 10 minutes at a flow rate of 200 slm. The results are discussed in 4.2.

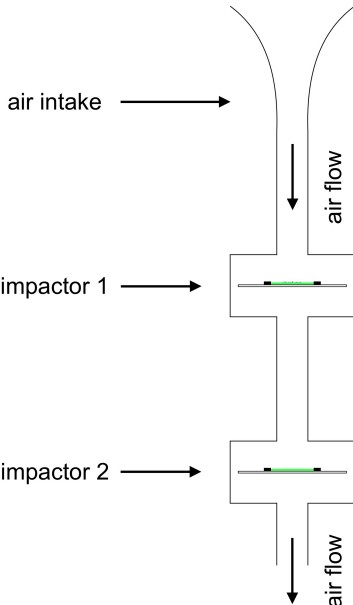

**Figure 7.** Schematic sketch of the extraction efficiency experiment with two identical impactors (impactor 1 and impactor 2) connected in a cascade configuration to investigate particle extraction efficiency. At 100 % efficiency, all particles would be extracted by impactor 1, leaving no particle for impactor 2.

## 3.3 Potential particle contamination of the sample carrier (Contamination Test)

Upon analysing the sample carrier using an optical microscope, it cannot be distinguished whether the particles on the sample carrier were collected during the airborne particle collection operation or inadvertently by contamination before or after the sampling operation. By using a laminar air flow box as previously described, contamination during manufacture and storage can be reliably prevented. And with the experiments described in the following, it was examined whether and, if so, what number of particles were inadvertently applied to the sample carrier by the handling of the sample carrier on ground at the site of operation as well as during the ascent and descent of the multicopter UAV.

### 3.3.1 Contamination on ground

At the site of operation, the particle sample carrier is exposed to atmospheric air during installation in and removal from the impactor. This exposure usually lasts less than 30 seconds, but nevertheless could lead to a contamination of the sample carrier with particles, in particular if the particle concentration in the ambient air

is exceptionally high. In a first investigation carried out in the afternoon (2:15 to 2:30 p.m. local time) of March 10, 2017 at the airfield in Poltringen, a sample carrier was removed from its protective packaging and exposed to ambient air for 15 minutes on the roof of a car about 1.8 meters a.g.l.. The sample carrier was then re-packaged and transported to the laboratory where it was treated and sealed in a particle free laminar air flow box to prevent any further contamination. The results are discussed in 4.3.

### 3.3.2 Contamination during ascent and descent

As observed during the smoke plume tests, an inflow of air into the air intake of the PCS appears during the hovering flight of the multicopter UAV even if the blower of the particle collection system is switched off. It is expected that this inflow incorporates aerosol particles onto the sample carrier and thus has to be regarded as a potential source of contamination. During vertical ascent of the multicopter UAV with a typical speed of 6 m s$^{-1}$ and the correspondingly higher propeller power, this effect is likely to be even more pronounced. Therefore, an experiment was carried out in the afternoon of March 10, 2017. A flight was carried out with the fully equipped multicopter UAV but with the blower of the PCS remained switched off. Upon start, the multicopter UAV climbed up at maximum ascent speed to an altitude of 300 m a.g.l.. After one minute of hovering, the multicopter UAV was descended to 50 m a.g.l., followed by a new ascent with maximum climb rate to an altitude of 200 m a.g.l.. After one minute of hovering the multicopter UAV was descended to ground and landed. Then the sample was transported to the laboratory where it was treated and sealed in the laminar air flow box as described earlier. In total, 450 m of ascent and descent in about 2.5 minutes were performed plus 2 minutes hovering time. The results are discussed in 4.3.

### 3.4 Aerosol particle collection flights

Numerous aerosol particle collection flights were carried out in March 2017 to evaluate the scientific potential of a multicopter UAV equipped with the newly developed PCS. The major aim of developing such a PCS was the collection of aerosol particles at different altitudes and their quantitative determination. For the present study we focused at first on the quantitative determination of the concentration of pollen grains. The airfield in Poltringen near Tübingen in Germany was chosen as launch site with regard to an existing official flight permit for UAV flights up to an altitude of 300 m a.g.l.. The airfield is located on an elevated plain above the Ammer Valley that is intensely used for agriculture. The site is about 2 km away from the 150 km$^2$ large Schönbuch Forest, a natural reserve, mainly consisting of a mixed deciduous and coniferous forest extending to the NE and forming an escarpment in the landscape arising about 70 m from the basal plain.

Three series of aerosol particles collection flights were carried out on March 3, 10, and 16, 2017, at the airfield in Poltringen with three flights each day. Table 1 gives an overview of these aerosol particle collection flights including data concerning the hovering altitude above ground level, at which the blower of the particle collection system was activated, the airborne particle collection start time, as well as the

measured air temperature, wind direction, and wind speed on ground. On March 3, 2017, the blower of the PCS was activated during hovering in 25 m, 100 m, and 200 m altitude a.g.l. and also – as an additional measurement – on ground with the propellers of the multicopter UAV being not in operation. On March 10 and 16, 2017, the PCS was activated during flights in 25 m, 200 m, and 300 m altitude a.g.l. The particle

5   collection duration at each altitude was 10 minutes, with a sampled air volume of 2,000 standard litres, corresponding to 2 m$^3$ under standard conditions, which are 20°C and 1,013.25 hPa according to the data sheet of the mass flow sensor.

Prior to each day of aerosol particle collection flights, the bellmouth-shaped air inlet, the tube leading to the impactor, the O ring and the two housing halves were cleaned in an ultrasonic bath with soapy water for 15

10  min, then rinsed with deionized and filtered (0.3 µm Membrane Filter) water and dried in a particle-free environment (laminar air flow box, HEPA H14 filter). Once the parts were dried, the impactor was assembled (excluding sample carrier) and packed together with the inlet into a new, clean, sealable storage bag.

In the field again, shortly before the particle collection operation, the impactor was taken out of the sealed

15  storage bag, the sample carrier was installed in the impactor and the inlet was plugged onto the tube leading to the impactor. In-between the sampling flights shortly before the next flight operation and shortly before installation of the next unloaded sample carrier, the impactor and the bellmouth-shaped inlet were flushed with filtered air using an battery operated electric blower with a medical ventilation filter installed on its inlet (type Pall Ultipor 100, > 99.999 % retention of airborne bacteria and viruses).

20  The sample carriers were treated post-flight as described previously. Identification and counting of the collected particles were visually performed using the Olympus transmitted light microscope BX50 at 400 times magnification. The entire area of the slides was counted row by row. Identification was assisted by a reference collection and literature by Beug (2004). The results are discussed in 4.4.

**Table 1.** Aerosol particle collection flights carried out on March 3, 10, and 16, 2017 performed in different hovering altitudes.

| Date | altitude a.g.l. | start time of particle collection | air temperature on ground | wind direction on ground | wind speed on ground |
|---|---|---|---|---|---|
| March 3, 2017 | multicopter on ground | 15:55 | | | |
| March 3, 2017 | 25 m | 15:20 | | | |
| March 3, 2017 | 100 m | 15:05 | 15 °C | NO | 1 m s$^{-1}$ |
| March 3, 2017 | 200 m | 15:44 | | | |
| March 10, 2017 | 25 m | 14:57 | | | |
| March 10, 2017 | 200 m | 14:38 | 17 °C | O | 1,8 m s$^{-1}$ |
| March 10, 2017 | 300 m | 15:40 | | | |
| March 16, 2017 | 25 m | 14:18 | 19 °C | S | 1,9 m s$^{-1}$ |
| March 16, 2017 | 200 m | 13:55 | 19 °C | S | 1,8 m s$^{-1}$ |
| March 16, 2017 | 300 m | 14:40 | 20 °C | W | |

# 4 Results and discussion

## 4.1 Position of the air inlet with regard to isokinetic sampling (Smoke Plume Test results)

The "Smoke Plume Tests" allow a quantitative determination of the air flow velocities. Despite their limited resolution, the results obtained here are in good agreement with the CFD calculations reported by Haas et al.
10  (2014): The smoke plume approaching 20 cm above the propellers of the multicopter UAV is directly captured by the propellers (Fig. 6B, middle smoke plume). Also the smoke plume approaching 80 cm above the multicopter UAV is strongly affected and accelerated downwards (Fig. 6A, lower smoke plume). The smoke plume approaching 1.8 m above the multicopter UAV, on the other hand, is already only very slightly affected (Fig. 6B, upper smoke plume). And the smoke plume approaching 2.4 m above the multicopter
15  UAV remains unaffected (Fig. 6A, middle smoke plume). Thus, these results correspond very well with the CFD-calculations reported by Haas et al. (2014), according to which the air volume mixed by the propellers of the multicopter UAV extends only about 2 m above the multicopter UAV. In addition, Fig. 6B also shows that the air volume mixed by the propellers extends further below the multicopter UAV than above the multicopter UAV, as predicted by the CFD-calculations.

With regard to the isokinetic sampling conditions concerning the *direction* of the air flow velocity vectors, it was observed that a plume of smoke approaching (due to prevailing side wind) horizontally 50 cm above the propellers of the multicopter UAV is caught by the downwash produced by the propellers and accelerated vertically downwards. When the smoke plume reaches the propellers, it is completely deflected from the original horizontal flow direction into a vertical flow direction. Already 30 cm above the propellers, the smoke plume is deflected in the vertical direction to an extent that it encloses an angle of about 20° with the vertical direction. As a result of these observations it was decided to orient the air intake of the PCS vertically upward and to position its open end 30 cm above the propellers of the multicopter UAV. In this position, the bell mouth shape of the air intake of the PCS enables substantially isokinetic sampling with regard to the direction of the air flow velocity vectors at least during hovering mode of the multicopter UAV and with side winds of less than 3 m/s, independent of the direction of the side wind.

With regard to the isokinetic sampling conditions concerning the *magnitude* of the velocity vectors, successive frames of the video sequences recorded during the visual air flow tests were evaluated. A horizontally approaching smoke plume begins to deflect in a vertical direction. Within three frames of the recorded video sequences, corresponding to 0.12 s, characteristic sections of the smoke plume cover a vertical distance between 15 and 20 cm, thus vertically arriving at a level about 30 cm above the propellers of the multicopter UAV where the air intake of the PCS is positioned. Under the simplified assumption of a uniform vertical acceleration, the vertical velocity component at this level can be calculated to be between 2.5 and 3.3 m s$^{-1}$. As the assumption of a uniform vertical acceleration is probably a strong simplification of the actual circumstances, a more precise determination of the vertical acceleration and velocity of the air flow above the multicopter UAV would be a valuable aspect of future work on this subject.

The circular opening width of the free (wider) end of the bell mouth shaped air intake has an inner diameter of 69 mm (Fig. 2). Thus, at an air volume flow of 200 litres per minute, the average flow velocity is about 0.9 m s$^{-1}$. Since it has to be assumed that the air flow velocity in the edge region of the bell mouth shaped air intake is significantly lower than in its centre region, the flow velocity in the centre region is to be expected above the average value of 0.9 m s$^{-1}$. This value is probably still less than the previously estimated vertical velocity component of the air to be drawn-in. Thus, despite of the high air volume flow of 200 litres per minute drawn in, a somewhat sub-isokinetic sampling is to be assumed with regard to the magnitude of velocity vectors. If necessary, the opening width of the free end of the bell mouth shaped air intake can be varied for future sampling operations to even better match the isokinetic sampling conditions.

As a result, positioning the air intake of the PCS 30 cm above the propellers of the multicopter UAV in combination with the vertically oriented and appropriately dimensioned bell mouth shaped air inlet ensures substantially isokinetic sampling conditions at high air volume flows of 200 litres per minute, even – within certain limits – regardless of prevailing side wind direction and speed.

## 4.2 Extraction efficiency of the impactor (Cascade Test results)

The extraction efficiency of the impactor was determined by visual analysis of sample carriers of two identical impactors connected in cascade and through flowed by the same air flow as shown schematically in Figure 7. At an ideal extraction efficiency of 100 %, all particles would be extracted by impactor 1 and thus no particles would be deposited on the sample carrier of impactor 2. The results of the visual analysis are shown in Table 2.

**Table 2.** Number of pollen grains collected in impactor 1 and impactor 2 of the arrangement of Figure 7 for determination of the retention rate and thus the extraction efficiency of the newly developed impactor.

| | Number of counted pollen grains on impactor stage $N_1$ and $N_2$ | | |
|---|---|---|---|
| genus | upstream impactor $N_1$ | downstream impactor $N_2$ | retention rate R= $N_1/(N_1+N_2)$ |
| *Taxus* | 806 | 3 | 99.63 % |
| *Alnus* | 194 | 2 | 98.97 % |
| *Corylus* | 49 | 1 | 97.96 % |
| *Pinus* | 1 | 0 | 100.00 % |
| **total** | **1050** | **6** | **99.43 %** |

The particle extraction and retention capability of the newly developed PCS was demonstrated for pollen of the genera *Taxus*, *Alnus*, and – with restrictions concerning statistical data base – *Corylus* and *Pinus*, which were present in the air at the time of the extraction efficiency experiment. While the number of pollen grains of the genera *Corylus and Pinus* are regarded of being too small for a statistical evaluation, the number of pollen grains of the genus *Taxus* and *Alnus* collected in upstream impactor no. 1 were about 100 to 250 times the number of corresponding particles in downstream impactor no. 2. As a result, the extraction efficiency, or retention ratio, of the impactor under the given conditions (200 litres per minute) concerning the pollen grains of genera *Taxus*, *Alnus*, and *Corylus* is at least 98 %.

With regard to the question whether this high extraction and retention rate also applies to other particles, i.e. to smaller particles, it should be noted that in the widely used Burkard pollen trap a mean jet velocity of 6 m s$^{-1}$ is sufficient enough to reliably extract pollen grains and spores from the air. For the widely used Burkard pollen traps, a modified orifice with a reduced width of 0.5 mm is available, which increases the mean jet velocity to 24 m s$^{-1}$ in order to improve the trapping efficiency for particles in the range 1-10 μm diameter (Datasheet Burkard 7 Day Recording Volumetric Spore Sampler, Burkard Scientific). As shown in Figure 9, the newly developed impactor (working with a mean jet velocity of 50 m s$^{-1}$) extracts aerosol particles having a size between the resolution limit of the light microscope (being in the range of 1 μm) and

approximately 60 µm. Further investigations are necessary to check whether the high extraction rates (of at least 98 %) determined for pollen of the genera Taxus, Alnus, and Corylus (with a typical size between 20 and 30 µm) also apply to particles in the µm and sub µm range.

### 4.3 Measurement errors and particle contamination (Contamination Test results)

5  The PCS, the visual identification, and the counting of particles are subject to various influences, which potentially form a source of errors with regard to the determination of the actual concentration of particles in the ambient air. An overview of these influences at the different components of the PCS, namely air intake, impactor, and mass flow sensor is given in Figure 8.

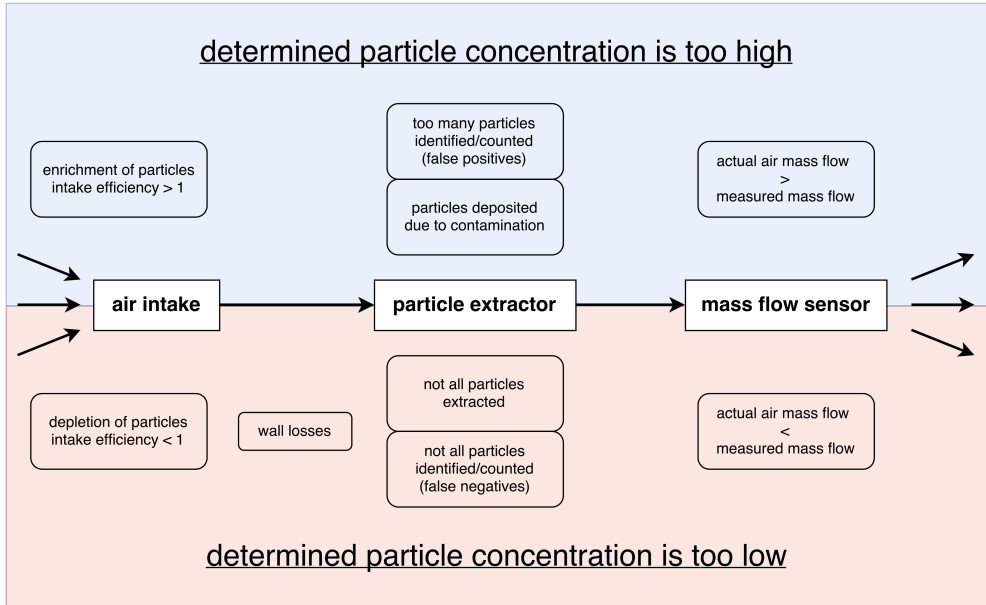

10  **Figure 8.** Overview of the possible influences of the different components of the newly developed particle collection system (PCS) on the finally determined particle concentration. The components of the PCS, at which the influences can occur, namely air intake, impactor, and mass flow sensor, are arranged along the horizontal axis. Influences that can lead to the determination of a particle concentration higher than the actual particle concentration are shown in the upper half of the figure (blue background), whereas the influences that can lead to the determination of a particle concentration lower than the actual particle concentration are shown in the 15  lower half of the figure (red background).

The first source of measurement error might occur during the air intake. If the ambient air is not drawn-in under isokinetic conditions, i.e. with the same velocity (by magnitude and direction) as the air approaching the air intake, then the air drawn in might be enriched or depleted with particles due to mass inertia effects. The multicopter UAV air flow tests have shown that by the suitable placement and design of the bell mouth

shaped air intake in combination with the operation of the PCS onboard the multicopter UAV in hovering flight mode result in almost isokinetic sampling conditions provided there are no excessive side winds. In order to be able to give an estimate of the error caused by non-100 % ideal isokinetic sampling, further investigations are required. A loss of particles, which have been already drawn-in, could occur due to adhesion to the wall of the air intake as well as to the wall of the downstream connecting pipes ('wall losses', Fig. 8). It is expected that such wall losses are of minor importance for the newly developed PCS with regard to its high air stream velocity of about 50 m s$^{-1}$ in the connecting pipe upstream of the impactor. In the impactor itself, an incomplete extraction of the particles would lead to a too low number of particles deposited on the sample carrier. However, according to the experiments performed within the scope of this study, the particle extraction rate of the impactor is at least 98 % for pollen grains.

Particle contamination is a potential error source that leads to higher particle numbers deposited on the sample carrier. Within the present study, experiments concerning potential contamination on ground as well as particle contamination during ascent and descent of the multicopter UAV were performed. Concerning the potential particle contamination on ground, in total 4 pollen grains were identified on the sample carrier, i.e. 2 of the genus *Taxus*, 1 of the genus *Alnus*, and 1 of the genus *Corylus*, as the result of a 15 minutes exposure of the uncovered sampling carrier to the ambient air. This small number is certainly also due to the lack of local sources such as pollinating trees or bushes within a radius of 150 m around the location of exposure (airfield in Poltringen).

For the evaluation of these results, the concentration of the pollen grains in the ambient air must be taken into account. The contamination experiments were carried out on March 10, 2017 at the same time as the aerosol particle collection flights. The mean values of the concentrations measured at the three altitudes (25 m, 200 m, and 300 m a.g.l.) are: 53 pollen grains per m$^3$ of the genus *Taxus,* 44 pollen grains per m$^3$ of the genus *Alnus*, and 16 pollen grains per m$^3$ of the genus *Corylus* (Tab. 3). Thus, the contamination during the exposure of the sample carrier for 15 minutes on ground represents between 3 % and 6 % of the number of pollen particles in one cubic meter of ambient air. With regard to the fact that the sample carrier is exposed to ambient air for handling purposes usually for less than 30 seconds, a contamination of 0.1-0.2 % is expected, which is negligible for most applications. This small particle contamination on ground can be further reduced or even excluded by employing a mobile laminar air flow box in the field. Furthermore, the lateral position of the particles deposited on the gelatine surface of the sample carrier enables an appraisal whether the particles were deposited during sampling or being the result of contamination on ground: while particles deposited during the sampling operation are within a circle corresponding to the contour of the open jet, particles deposited by contamination on ground are statistically distributed over the entire surface.

More relevant is the contamination of the particle sample carrier during ascent and descent of the multicopter UAV. During the corresponding contamination experiment, 450 m of ascent and descent were performed within 2.5 minutes, and in addition 2 minutes of hovering in 200 m and 300 m altitude a.g.l.. In total 17 pollen grains, 8 of the genus *Taxus*, 6 of the genus *Alnus*, and 3 of the genus *Corylus*, were

identified on the sample carrier. As a result, the number of pollen grains deposited on the sample carrier during ascent, hovering, and descent represents between 15-19 % of the number of pollen in one cubic meter of ambient air. If, for simplification, the contamination during hovering is neglected, then a contamination of 3-4 % for every 100 m ascent and descent is caused. As a result, relevant contamination of the particle sample carrier may occur during ascent and descent of the multicopter UAV. The extent of the contamination depends on the altitude the multicopter UAV is elevated to, and also depends to the particle concentration in the layers of air crossed by the multicopter UAV during ascent and descent.

During the visual identification and counting of the particles, it is possible that contrast differences when using the transmitted light microscope are erroneously identified as particles (false positives) and/or that some particles are double-counted. Furthermore, it is possible that some particles are not or not correctly identified (false negative) and/or that some particles are overlooked. This potential source of error was excluded in the present study by entrusting particularly experienced scientists with the visual identification and counting of the particles, which still is the golden standard for pollen concentration measurement (Oteros et al., 2015).

Finally, a potential error source exists with regard to the accuracy of the mass flow sensor SFM3000-200-C. It is evident that any difference between the actual and measured air mass flow produces a corresponding error in the determined particle concentration. According to the data sheet of the mass flow sensor, within the temperature range of –20 °C to + 80 °C, the error is typically 1.5 %, maximum 2.5 %, of the measured value.

### 4.4 Airborne particle collection operation

### 4.4.1 Results of the aerosol particle collection flights

The number of particles collected during the aerosol particle collection flights on March 3, 10, and 16, 2017 from 2 m$^3$ of sampled air and subsequently counted by visual microscopic analysis of the respective sample carriers are summarized in Table 3.

**Table 3.** Summary of the number of collected particles (from 2 m$^3$ sampled air, respectively) using the new particle collection system (PCS) onboard the multicopter UAV during the aerosol particle collection flights carried out in March 2017 (top); in addition, the comparison of these measured values with the forecast data of the Deutscher Polleninformationsdienst (PID) (middle); and the pollen concentrations measured by MeteoSwiss measured by a commercially available Burkard pollen sampler in Zurich (bottom);

**Results of the Measurements performed at Poltringen Airfield with the newly developed Particle Collection System mounted on the multicopter UAV**

|  | March 3 | | | | March 10 | | | March 16 | | |
|---|---|---|---|---|---|---|---|---|---|---|
| collection start time (local time) | 15:55 | 15:20 | 15:05 | 15:44 | 14:57 | 14:38 | 15:40 | 14:18 | 13:55 | 14:40 |
| flight altitude (in m a.g.l) | ground | 25 m | 100 m | 200 m | 25 m | 200 m | 300 m | 25 m | 200 m | 300 m |
| *Taxus* | 32 | 22 | 24 | 2 | 113 | 133 | 70 | 135 | 175 | 88 |
| *Corylus* | 27 | 35 | 30 | 29 | 32 | 36 | 26 | 4 | 1 | - |
| *Alnus* | 128 | 167 | 159 | 181 | 109 | 91 | 63 | 18 | 11 | 12 |
| *Cyperaceae* | - | - | - | - | 5 | 2 | 2 | - | - | - |
| *Salix* | - | 3 | 2 | 1 | 9 | 3 | 5 | 23 | 3 | 10 |
| fungal spores type 1 | 22 | 5 | 17 | 2 | 200 | 114 | 131 | 2 | 2 | 2 |
| fungal spores type 2 | 16 | 1 | 3 | 4 | 3 | 4 | 4 | 2 | 5 | 3 |
| opaque particles >20 μm | 2 | 11 | 4 | 9 | 52 | 33 | 26 | 30 | 34 | 16 |

**Comparison to the Statement of the Deutscher Polleninnformationsdienst (PID)**

| | Statement of the Deutscher Polleninformationsdienst (PID) "Wochenpollenvorhersage" | | |
|---|---|---|---|
| | Week of March 1, 2017 (KW9) | Week of March 8, 2017 (KW10) | Week of March 15 (KW 11) |
| Pollen of the genera *Taxus* | first weak load "erste schwache Belastung" | short time large amount "kurze Zeit große Menge" | the most abundant genus of Pollen "die mengenmaäßig haäufigste Pollenart" |
| Pollen of the genera *Corylus* and *Alnus* | first high concentration "erstmals hohe Konzentration" | approaches the end "nähert sich dem Ende" | fadet ("abgeblüht") |

**Comparison to the Measurements of MeteoSchweiz performed in Zürich**

| | March 3, 3017 | March 10, 2017 | March 16, 2017 |
|---|---|---|---|
| *Alnus* | Number of pollen grains per m$^3$ | | |
| PCS in Poltringen (25 m a.g.l.) | 18 | 15 | 2 |
| Burkard sampler in Zürich | 41 | 20 | 5 |
| *Corylus* | Number of pollen grains per m$^3$ | | |
| PCS in Poltringen (25 m a.g.l.) | 84 | 55 | 9 |
| Burkard sampler in Zürich | 39 | 45 | 8 |

Only pollen of the genus *Taxus*, *Corylus*, *Alnus*, *Cyperaceae*, and *Salix* were counted and listed as well as two types of fungal spores. Fungal spore type 1 probably belongs to the genus *Cladosporium*, whereas fungal spore type 2 most likely belongs to the genus *Epicoccum*. Furthermore, large opaque particles with a longitudinal extension of more than 20 μm were counted; many of these particles having a wood fibre-like structure and the appearance of residues of burned wood or charcoal. Additionally, a large number of small aerosol particles down to a size of less than 1 μm were visible under the microscope, but are not listed as they cannot be reliably identified by visual inspection only. Figure 9 shows a photograph of the sample carrier content as an example of one of the collection flights.

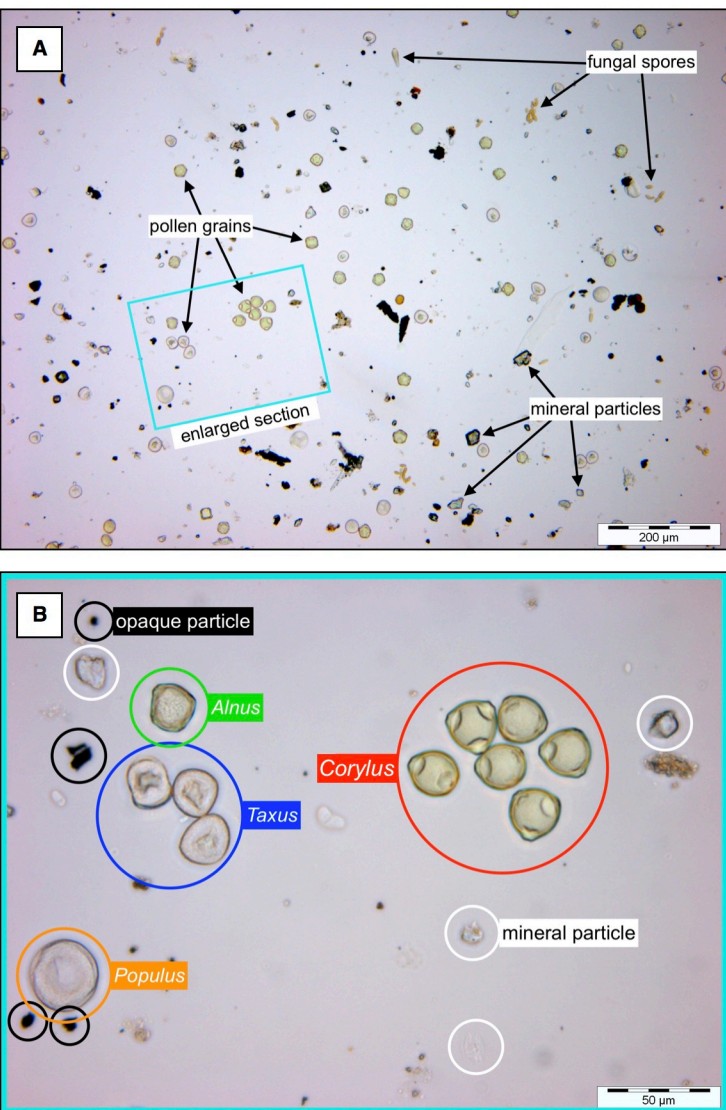

**Figure 9.** (A) Microscope photograph of a sample carrier loaded with various aerosol particles deposited during a multicopter UAV collection flight at an altitude of 300 m above ground level. The section bounded by the cyan rectangle in (A) is shown enlarged in (B). (B) Enlargement shows clusters of *Corylus* and *Taxus* pollen grains as well as transparent mineral and opaque particles in various sizes.

The amount of collected pollen grains, fungal spores, and large (> 20 µm) opaque particles vary significantly between the three sampling days as well as within each sampling day with the respective sampling altitude a.g.l.. Generally, the results reflect the expected type and concentration of pollen usual for this season (Fig. 10).

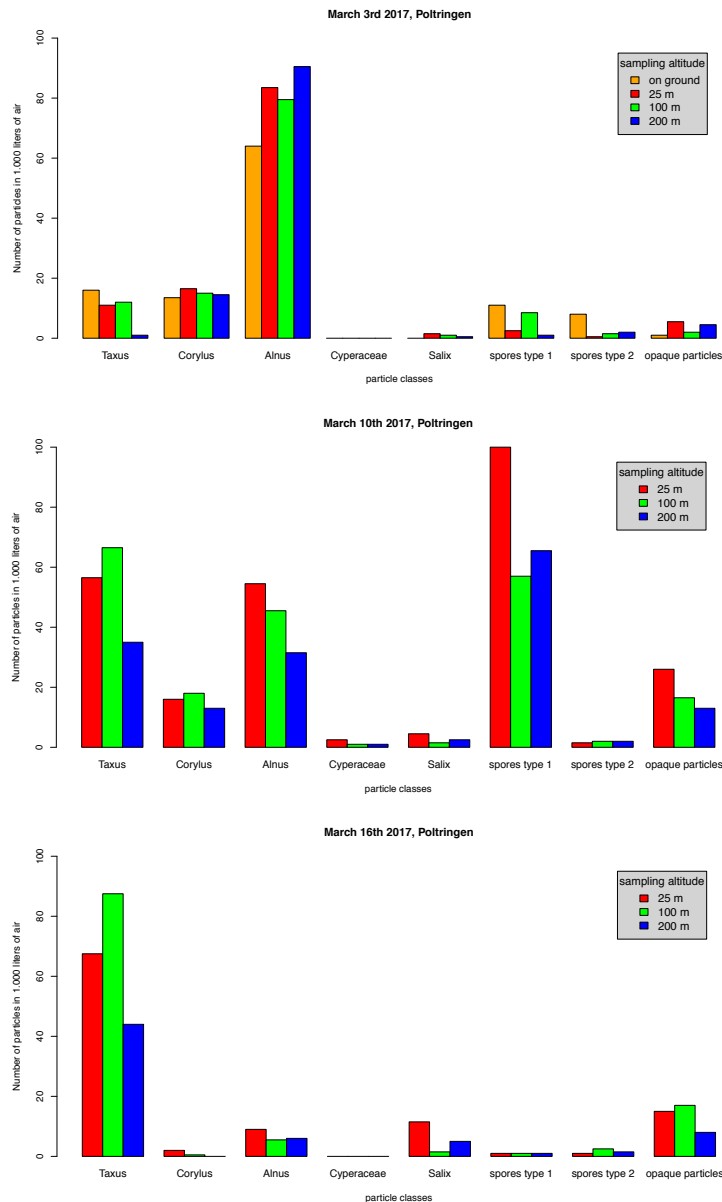

**Figure 10.** Graphical representation of the measured concentrations of particles (in particles per m$^3$ of sampled air) collected during
the aerosol particle collection flights on March 3, 10, and 16, 2017. Colour differences in the individual bars represent the particle
concentration at different altitudes. It should be noted that only on March 3, 2017 a sampling operation on ground has been carried
out with the propellers of the multicopter UAV being switched off. On that date, sampling operations have been carried out also in

altitudes of 25 m, 100 m, and 200 m above ground level (a.g.l.), whereas on March 10 and 16, 2017 sampling operations have been carried out in altitudes of 25 m, 200 m, and 300 m a.g.l., respectively.

Only the numbers of the pollen of the genera *Taxus*, *Corylus*, and *Alnus* as well as large (> 20 μm) opaque particles were high enough (i.e. more than 10 particles per m$^3$) to allow a reliable statistic evaluation. Pollen of the genus *Salix* appeared only in small numbers during all three sampling days, and pollen of the genus *Cyperaceae* even were collected solely on March 10, 2017 at all. Fungal spores of type 1 and 2 occurred on all three sampling days only in small numbers, except of March 10, 2017, when the fungal spores of type 1 were collected in a remarkable large number.

For all sampling altitudes, the concentration of pollen of the genus *Taxus* increased in the period between March 3 and March 16. For example, the concentration value measured at an altitude of 25 m a.g.l. rose from 11 pollen grains per m$^3$ on March 3, to 57 pollen grains per m$^3$ on March 10, and finally to 68 pollen grains per m$^3$ on March 16. Contrary to that, the concentration of pollen of the genus *Alnus* at an altitude of 25 m a.g.l. decreased from 84 pollen grains per m$^3$ on March 3, to 55 pollen grains per m$^3$ on March 10, and finally to 9 pollen grains per m$^3$ on March 16. The concentration of pollen of the genus *Corylus* measured on March 3 and 10 remained almost constant, but decreased significantly on March 16. For example, at an altitude of 25 m a.g.l., 18 pollen grains per m$^3$ were counted on March 3, 16 pollen grains per m$^3$ on March 10, but only 2 pollen grains per m$^3$ on March 16. Spores of type 1 and 2 were collected in consistently small numbers of less than 10 spores per m$^3$ in the period between March 3 and March 16. One exception appeared on March 10 when the concentration values of spores of type 1 reached more than 50 spores per m$^3$ in all three sampling altitudes (Fig. 10).

For many of the pollen genera collected during the particle collection flights in March 2017, the pollen grain concentrations measured in altitudes of 100 m, 200 m, and 300 m a.g.l. are in the same order of magnitude as the pollen grain concentration measured near to the ground (25 m). This applies in particular to the pollen genera detected in a large number during the measuring flights. One possible explanation for this observation is that all particle collection flights were carried out in the afternoon between 2 pm and 4 pm local time during early spring days with relatively high number of sunshine hours and no rain. It can be therefore assumed that on each of the three days a convective boundary layer had formed comprising of mixed air and thus homogenizing the concentration of the aerosol particles. This mixing process takes place within the entire convective boundary layer usually extending up to an altitude of 1,000 to 2,000 m a.g.l. in the afternoon (Stull, 2012). It also can be concluded that the sources of the collected pollen were not only local, rather regional; otherwise a higher concentration would have been observed near the ground close to the local pollen source.

During the measuring flights on March 10 and 16, 2017, the concentration of pollen of the genus *Taxus*, which were the most frequently occurring pollen type at this time, was even higher at an altitude of 200 m a.g.l. than at 25 m a.g.l. When interpreting these results, it has to be kept in mind that the measuring flights at the different altitudes were carried out one shortly after the other and within a period of about 30 minutes,

but not concurrently. Thus, it cannot be completely ruled out that the higher pollen concentration at the altitude of 200 m a.g.l. is merely the result of a short-time change in the overall pollen concentration at the measuring site, for example due to gusting wind. On the other hand, it is remarkable that this phenomenon was observed both on March 10, when the concentration at 200 m a.g.l. was 18 % higher than at 25 m a.g.l., and on March 16, when the concentration at 200 m a.g.l. was even 30 % higher than at 25 m a.g.l..

The observation that the pollen grain concentration was higher at elevation than on ground is in good agreement with the results of Comtois et al. (2000) who conducted pollen concentration measurements using a tethered balloon up to an altitude of 600 m a.g.l.. Their results revealed that the pollen grain concentration at 600 m a.g.l. can be similar or, depending on the pollen genus, even higher than on ground. Also Damialis et al. (2017) reported recently a higher pollen concentration even at an altitude of 2,000 m a.g.l. compared to the values measured on ground.

During the measuring flights on March 10, 2017 for both, the pollen of the genera *Taxus* and *Corylus*, the highest pollen concentration values were measured at the altitude of 200 m a.g.l., whereas for pollen of the genus *Alnus* the highest pollen concentration values were measured at the altitude of 25 m a.g.l.. This might be an indication that the transport mechanisms and corresponding transport parameters are significantly specific to the respective pollen genus, even possibly resulting in the transport of pollen at genus-specific circumstances and altitudes. In order to gain in-depth knowledge on this topic, further experiments are necessary, such as concurrent measurements of pollen concentrations in different altitudes.

During the measuring flights on March 3, 2017, in addition to the aerial sampling at various altitudes, one sample was taken on the ground with the propellers of the multicopter UAV switched off and only the blower of the PCS being activated. The concentrations of the most frequently occurring pollen of the genera *Corylus* and *Alnus* were 23 % lower than at the altitude of 25 m a.g.l.. This might be an indication that sedimentation or filtration of the pollen grains by ground-level vegetation leads to a depletion of the pollen concentration in ground-near air layers. Another possible explanation for this observation is, that the inflow occurring at the air intake of the PCS is increased due to the operation of the propellers of the multicopter UAV during the aerial sampling, and thus the intake capture efficiency of the PCS might be increased, for example as a result of sub-isokinetic sampling conditions. If this is the case, and if this effect is reproducible, which requires further experiments, then such an increase of intake capture efficiency of the PCS could be used advantageously, since this would allow a further reduction in the sampling period necessary to collect a predetermined amount of aerosol particles.

### 4.4.2 Comparison to pollen forecast information services

The PID publishes and stores online (http://www.pollenstiftung.de/aktuelles/) weekly forecasts on the development of the pollen concentration in Germany, especially for pollen genera with a known allergy risk. The comparisons of the forecasts with the values measured with the newly developed PCS onboard the multicopter UAV are shown in Table 3. The pollen concentration of genus *Taxus* measured with the PCS

rose over the three sampling days, for example at an altitude of 25 m a.g.l. from 11 to 68 pollen grains per m$^3$. This is in agreement with the PID forecast that also predicted a significant increase in the pollen concentration of *Taxus* for this period. The agreement of the PCS measurements with the PID forecasts is also reflected in other measured pollen concentrations such as the genera *Corylus* and *Alnus*. As predicted by the PID we also measured a significant decrease in the pollen concentrations from 18 towards 2 (genus *Corylus*) and from 84 towards 9 (genus *Alnus*). The good agreement between the forecasts of the PID and the results of the particle collection flights conducted in this study is a first strong indication that the newly developed PCS reliably determines the pollen concentration in ambient air, even when operated onboard of an airborne multicopter UAV.

The allergy centre of Switzerland (Allergiezentrum Schweiz) provides online not only forecast information on expected, but also of the actual daily pollen concentration. These accurate data are provided from a network of 14 measuring stations equipped with BURKARD pollen traps that are operated by MeteoSchweiz. For an evaluation (Table 3) of the pollen concentration values determined within our study, the measuring station of MeteoSchweiz in Zürich was selected. The selection is based on the relatively short distance of about 130 km between Zürich and our measuring site in Poltringen, an almost identical altitude a.s.l., and very similar temperature conditions during the measurement period (www.accuweather.com). Figure 11 shows the comparison of the pollen concentrations of the genera *Corylus* and *Alnus* measured on March 3, 10, and 16 with our PCS at an altitude of 25 m a.g.l. and by MeteoSchweiz using BURKARD pollen traps. On each of the three days, a generally slightly higher concentration of pollen of the genus *Corylus* was measured in Zürich than in Poltringen, but showing an almost parallel decreasing trend over the course of this period at both sites. In contrast, for pollen of the genus *Alnus*, a higher concentration was measured in Poltringen than in Zürich on each of the three days, but again showing an almost parallel decreasing trend towards the end of the sampling period. The slight differences in the absolute concentration values between the two sites might reflect the different dominating vegetation type in Poltringen and Zürich. In summary, it thus can be stated that the pollen concentration values determined during the measuring flights in Poltringen are in very good agreement with the corresponding pollen concentration values published by MeteoSchweiz.

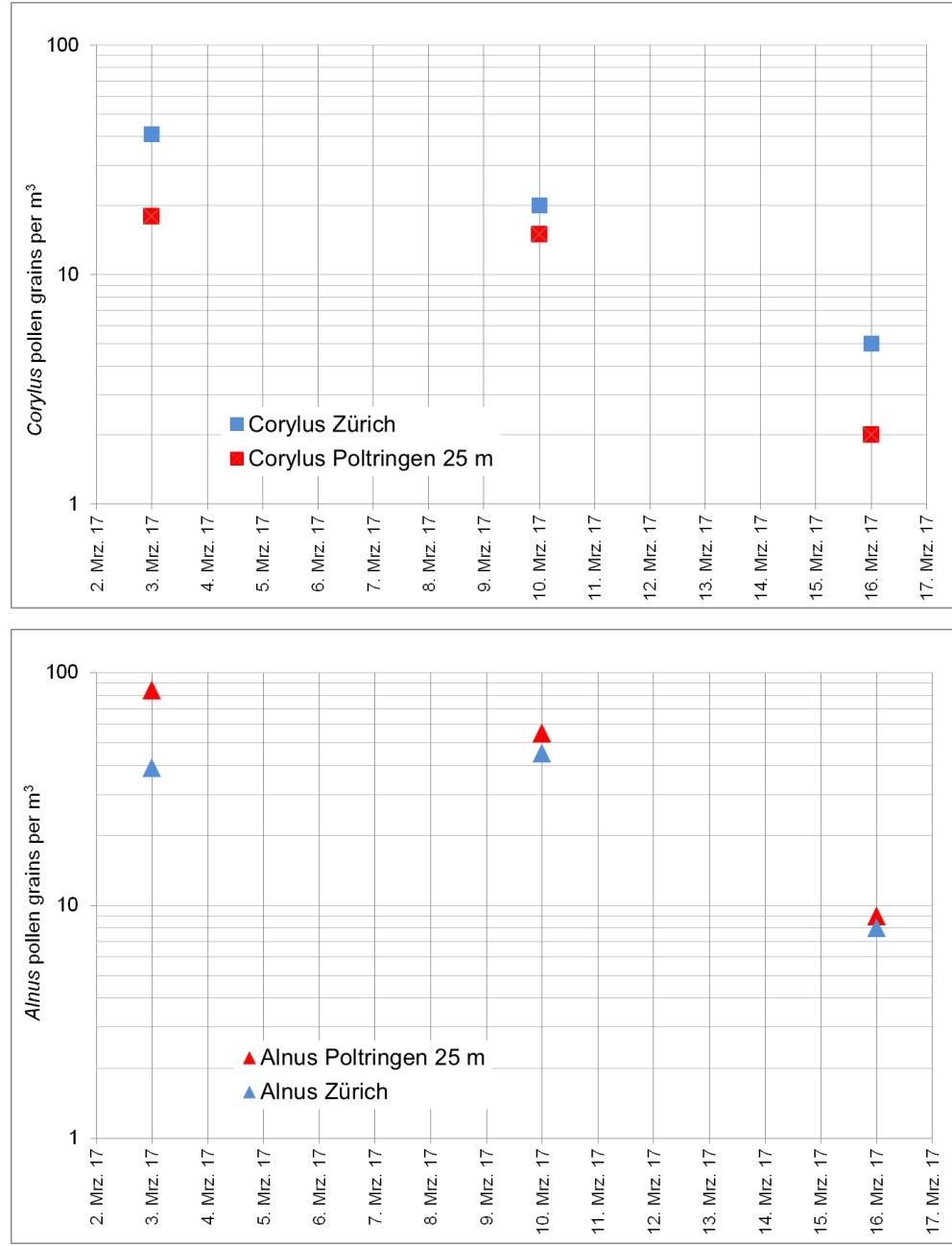

**Figure 11.** Concentration of pollen of the genus *Corylus* and *Alnus* collected in Poltringen with the new particle collection system
5 (PCS) operated onboard the multicopter UAV during hovering at 25 m a.g.l. on March 3, 10, and 16, 2017 in comparison with pollen

concentrations of the same genus published by MeteoSchweiz in Zürich measured by using a Burkard pollen sampler. The lowest altitude above ground level data for Poltringen are available from 25 m altitude a.g.l. for all three sampling days.

## 5 Conclusions

5    The presented multicopter based UAS with the newly developed impactor-based particle collection system (PCS) operated in-flight and onboard the multicopter UAV has proven to be a powerful and reliable system for aerosol particle collection in the ABL. More than thirty particle collection flights were carried out with this new UAS, each with a sampled air volume of 2 m$^3$ and at flight altitudes of up to 300 m a.g.l..

A particle separation efficiency of more than 98 % was determined for the newly developed impactor-based PCS despite the high air volume flow of 0.2 m$^3$ per minute. In order to achieve a high particle capturing 10    efficiency, the design and placement of the air intake was optimized by conducting and evaluating visual airflow tests. Easily interchangeable sample carriers guarantee a lean post-flight workflow with regard to visual analysis using transmitted light microscopy. The use of a laminar air flow box reliably protects the particle sample carriers from particle contamination during their manufacturing, handling, and storing.

Subject to a sufficiently high concentration of the corresponding particles in the air, the number of in-flight 15    collected particles was regularly well above one hundred during a ten-minute sampling operation. These large numbers of collected particles provide the possibility of reducing the volume of sampled air and thus reducing the aerial sampling period. Accordingly, particle collection flights at altitudes of up to 500 m a.g.l. and beyond are possible without any modification regarding the multicopter UAV.

The particle collection flights carried out during the pollen season in March 2017 at altitudes of 25 m, 100 m, 20    200 m and 300 m a.g.l. show remarkable vertical distribution of the various pollen genera and impressively illustrate the scientific potential of the newly developed PCS operated onboard a multicopter UAV, such as the determination and modelling of the propagation behaviour of pollen, spores and other airborne particles in the ABL (Aylor et al., 2006). In a more application-oriented context, it is very gratifying that the pollen concentration values measured with the new PCS onboard the multicopter UAV matches very well, both in 25    their absolute numbers as well as in their relative temporal change, with the pollen concentration predictions and pollen concentration data published by the two pollen information services Stiftung Deutscher Polleninformationsdienst (PID) and MeteoSchweiz.

## Acknowledgements

This research was financed through institutional funding of the Eberhard Karls Universität Tübingen. Martin Schön supported us in the manufacture of the bell mouth shaped air intake using FDM (Fused Depositing Modeling) three-dimensional printing technology. In addition, we would like to express our gratitude to Barbara Maier, Simone Schafflick and Wolfgang Kürner for their highly appreciated assistance in the implementation of numerous ideas into functional designs. In the realization of some technical solutions, we experienced kind support of companies established in the relevant technical fields; therefore we would like to express many thanks to Helmut Memmel and Alexander Post of the Daldrop + Dr. Ing. Huber GmbH + Co.KG in 72666 Neckartailfingen/Germany for the provided insights into cleanroom technology and the generous provision of high efficiency air filter elements, Manuel Meier of the Sensirion AG in 8712 Stäfa/Switzerland for a kind introduction in air mass flow measuring and the provision of sensor samples, Julia Ganter of the ebm-Papst GmbH + Co.KG in 78112 St. Georgen/Germany and Ernst Scherrer of the Micronel AG in 8317 Tagelswangen/Switzerland for the provision of miniature blower samples, and Dr. Jörg Haus and Rouven Möller of the Helmut Hund GmbH in 35580 Wetzlar/Germany as well as Matthias Werchan of Stiftung Deutscher Polleninformationsdienst for the provided insights into pollen and spores measurement and identification. Finally we would also like to thank the editor and the anonymous reviewer for their kind and highly qualified comments which have improved the quality of our manuscript.

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
