# Peer review of "A new multicopter based unmanned aerial system for pollen and spores collection in the atmospheric boundary layer"

_Atmospheric Measurement Techniques, 2018_

## Referee Comment (RC1) · Anonymous Referee #1 · 16 Nov 2018

The ms by Crazzolara et al. describes a new collection device onboard a commercial rotary-wing UAS to collect particles in the lower atmosphere. The authors report bioaerosol data from 9 different flights conducted over three days in March, 2017. They also did some rather crude examinations of airflow over the device using colored smoke emitted from a pole at three different heights. They attempted to quantify the collection efficiency of the sampling device by putting two of the trapping surfaces inline for a single sampling interval. The manuscript has numerous formatting issues with the figures and tables (these are not even close to being ready for publication), and needs a major overhaul. There appear to be some errors with how fungi were identified (labeled fungal spores in one of the figures do not match the genera listed in the manuscript).

[Figure]

Some key references are missing from the manuscript regarding the sampling of fungi and pollen in the lower atmosphere with UAS. Comments below.

1. UAV should be replaced with UAS (unmanned aircraft system) throughout. Multi-copter should be replaced with rotary-wing UAS or hexacopter (the S900 is a hexacopter platform). Moreover, the platform itself is not new. It is the collection system being used on the UAS that is new and interesting. This needs to be re-shaped in the text.

2. The introduction and results and conclusions are missing some key references regarding the sampling of the lower atmosphere with UAS:

Aylor, D.E., Boehm, M.T. and Shields, E.J., 2006. Quantifying aerial concentrations of maize pollen in the atmospheric surface layer using remote-piloted airplanes and Lagrangian stochastic modeling. Journal of Applied Meteorology and Climatology, 45(7), pp.1003-1015.

Boehm, M.T., Aylor, D.E. and Shields, E.J., 2008. Maize pollen dispersal under convective conditions. Journal of Applied Meteorology and Climatology, 47(1), pp.291-307.

Gottwald, T.R. and Tedders, W.L., 1985. A spore and pollen trap for use on aerial remotely piloted vehicles. Phytopathology, 75(7), pp.801-807.

Hardin, P.J. and Hardin, T.J., 2010. Small scale remotely piloted vehicles in environmental research. Geography Compass, 4(9), pp.1297-1311.

Jimenez-Sanchez, C., Hanlon, R., Aho, K.A., Powers, C., Morris, C.E. and Schmale III, D.G., 2018. Diversity and ice nucleation activity of microorganisms collected with a small unmanned aircraft system (sUAS) in France and the United States. Frontiers in microbiology, 9.

Lin, B., Ross, S.D., Prussin II, A.J. and Schmale III, D.G., 2014. Seasonal associations and atmospheric transport distances of fungi in the genus Fusarium collected with unmanned aerial vehicles and ground-based sampling devices. Atmospheric environment, 94, pp.385-391.

Schmale III, D.G. and Ross, S.D., 2015. Highways in the sky: Scales of atmospheric transport of plant pathogens. Annual review of phytopathology, 53.

Schmale, D.G., Ross, S.D., Fetters, T.L., Tallapragada, P., Wood-Jones, A.K. and Dingus, B., 2012. Isolates of Fusarium graminearum collected 40–320 meters above ground level cause Fusarium head blight in wheat and produce trichothecene mycotoxins. Aerobiologia, 28(1), pp.1-11.

3. Details on the operation of the sampling device are limited. Was the device powered on remotely once the UAS had reached the desired altitude? Or was it sampling on its way up and down from the target sampling altitude? Is this why you conducted the profile missions? If not, why didn't include a remote switch to power a unit? In fact, you can use a light activated trigger sensor and turn the LEDs on and off from a DJI platform to act as a switch for this using the DJI platform and software.

4. What precautions were taken to clean the inlet and the inlet pipe in between sampling missions?

5. In general, the figure legends do not contain enough information for the figure to stand alone without referencing back to the text. Figure 5 is a great example of this.

6. What sort of quantitative data were measured for experiments described in Figure 6? There appear to be only qualitative observations. Could you use image-processing tools like IMageJ to formally track the plume of smoke? Did you trap any of the smoke particles on your collection device?

7. Your particle trapping efficiency experiments based on two inline trapping surfaces and a single experiment are just not enough. You should aerosolize known particle sizes (such as flourescent microspheres that you can buy at set size ranges), and attempt to trap them on your sampling device. Your efficiency will likely be linked to the size of your particle. Many of the smaller particles probably go cruising on by the

initial trapping surface. Your final inline sampler could be an impinger, to collect all of the material in a liquid and use that as a basis of quantification.

8. Table 1 needs to overhauled. Order by start time, not by altitude. Also, list stop time of collection. Why did the authors choose different sampling times on different days? What is the justification for this? Why not sample the same altitude at multiple times throughout a single day? As it stands, you only present 3 reps of data for 25m and 200 m. 100m is not replicated, and 300 m was only flown twice.

9. Delete Figure 8. This is really just meant for the discussion.

10. Table 3 needs to be completely overhauled. Consider separate rows for each flight, and separate columns for the pollen and fungi analyzed. I am concerned about the fungal genera presented in this table. The authors report Puccinia and Epicoccum, but the 'fungal spores' they show in Figure 9 do not appear to be representative of either of these genera.

11. Figure 10 needs to be formatted for publication. I'm not sure what the authors are trying to do here, since they show these data in Table 3.

12. I don't understand the need for Figure 11. Why was 25 m reported? Was this the altitude the reference data were recorded at?

13. Finally, no hypotheses are stated or tested. This makes it very difficult to judge the merits of this work. Did you expect to find different concentrations of pollen at different altitudes? If so, why? How might the concentrations of pollen change throughout a day or night? Did you hover at a single location? What about hovering a multiple locations, but maintaining precise altitudes? More flights are needed to really show the value of this platform. Do you know where the pollen is coming from? Just because a forest is nearby doesn't mean the pollen was coming from there...

---

## Author Comment (AC1) · 22 Jan 2019

**Response to comments of Referee #1**

We would like to thank Referee #1 for his valuable and thoughtful comments, which helped us a lot to improve the content and quality of our manuscript. In the following we have addressed all the comments of the Referee and incorporated changes in the manuscript as follows:

Blue: Comments of Referee #1

Black: Answers of Authors

*Black, italic, "":"Changes in the manuscript"*

**RC1: Anonymous Referee #1**

**1.1 UAV should be replaced with UAS (unmanned aircraft system) throughout.**

Actually the terms UAV and UAS are not used uniformly in literature; in particular, both terms are still common at present. Even in the key references kindly recommended by Referee #1, very similar aerial measurement systems are referred to as both "unmanned aerial vehicle (UAV)" (Schmale, 2010) as well as "unmanned aircraft system (UAS)" (Schmale, 2015).

In order to avoid conceptual ambiguities, we carefully use the terms UAV and UAS in different ways uniformly throughout the text: the term unmanned aerial vehicle (UAV) is used to describe the aircraft (as the aerial vehicle) itself, whereas the term unmanned aerial system (UAS) is used to describe the entire operation system, including the unmanned aerial vehicle, the payload with measuring equipment and further accessories to control the system from ground.

**1.2 Multicopter should be replaced with rotary-wing UAS or hexacopter (the S900 is a hexacopter platform).**

We deliberately choose the term "multicopter" because this term seems to us to be very accessible and is currently very common in the more recent relevant literature (e.g. the cited Brosy, 2017). The use of the term "rotary-wing" might be misleading with regard to the aerial vehicle actually applied during our work, since "rotary-wing" is – according to our understanding – also used as a generic term for helicopters.

On the other hand, the term "hexacopter" seems to be too narrow, since the number of propellers (e.g. four, six or eight) is of no relevance for the presented subject-matter.

With all due respect for the well-intentioned commentary, we would therefore prefer to keep the terms we have chosen.

1.3 Moreover, the platform itself is not new. It is the collection system being used on the UAS that is new and interesting. This needs to be re-shaped in the text.

We fully agree with your focusing on the particle collection system (PCS) as the new and interesting aspect. In order to further clarify this, we amended the abstract and the introduction by emphasizing that the PCS is "newly in-house developed" whereas the multicopter UAV is "commercially available" as already stated in the originally filed text on chapter 2.1., first line:

*Abstract: "The application of a newly in-house developed particle collection system (PCS) onboard a commercially available multicopter unmanned aerial vehicle (UAV) is presented as a new unmanned aerial system (UAS) approach for in-situ measurement of the concentration of aerosol particles such as pollen grains and spores in the atmospheric boundary layer (ABL)."*

*Introduction: "In this paper we present the structural design and first application of a newly in-house developed particle collection system (PCS) operated onboard a commercially available multicopter UAV (Fig. 1) for in situ measurement of the concentration of pollen and spores in the ABL"*

2. The introduction and results and conclusions are missing some key references regarding the sampling of the lower atmosphere with UAS.

Many thanks for the literature recommendations, emphasising especially the relevance of the topic in agriculture science.

As can be seen from the attachment no. 1, all references recommended by referee #1 have been introduced in the sections introduction or conclusions, and further have been added to the list of references (added references are highlighted).

3. Details on the operation of the sampling device are limited. Was the device powered on remotely once the UAS had reached the desired altitude? Or was it sampling on its way up and down from the target sampling altitude? Is this why you conducted the profile missions? If not, why didn't include a remote switch to power a unit? In fact, you can use a light activated trigger sensor and turn the LEDs on and off from a DJI platform to act as a switch for this using the DJI platform and software.

Yes, the newly developed particle collection system (PCS), namely its electrically operated blower, was powered on remotely once the UAS had reached the desired sampling altitude. We tried to make this clear e.g. with the last sentence in section 2.2.4 (page 10, line15: "One channel of the remote control system was used to switch the blower on and off when the multicopter UAV was airborne and the particle collection position was reached.") or with the second sentence of the second paragraph of chapter 3.4 (page 16, line 31: "Table 1 gives an overview of these aerosol particle collection flights including data concerning the altitude above ground level at which the particle collection system was activated,…").

Technical details concerning the remote operation of the blower are deliberately limited, because we had considered them to be too simple to mention and seemed superfluous to us in this context.

Nevertheless we apologize for any remaining ambiguities. To describe the procedure for particle collection even more clearly, we amended the text at the end of chapter 2.2.4 by adding the following to the text:

*"As long as the blower was switched on, i.e. as long as particle collection was performed, the multicopter UAV was maintained (by hovering) in this desired particle collection position. Before leaving this position, the blower was remotely switched off, thus terminating the particle collection operation. The value of the electrical current, drawn by the blower from the battery, was measured onboard the multicopter UAV and transmitted to and monitored on ground, to make sure that the blower has really went into operation (switched on) or was really out of operation (switched off)."*

Furthermore we have amended the following wording (underlined) in second paragraph of section 3.4:

"*Table 1 gives an overview of these aerosol particle collection flights including data concerning the hovering altitude above ground level, at which the blower of the particle collection system was activated, the airborne particle collection start time, as well as the measured air temperature, wind direction, and wind speed on ground. On March 3, 2017, the blower of the PCS was activated during hovering in 25 m, 100 m, and 200 m altitude a.g.l. and also – as an additional measurement – on ground with the propellers of the multicopter UAV being not in operation. On March 10 and 16, 2017, the PCS was activated during flights in 25 m, 200 m, and 300 m altitude a.g.l.. The particle collection duration at each altitude was 10 minutes, with a sampled air*

*volume of 2,000 standard litres, corresponding to 2 m$^3$ under standard conditions, which are 20°C and 1,013.25 hPa according to the data sheet of the mass flow sensor."*

**4. What precautions were taken to clean the inlet and the inlet pipe in between sampling missions?**

Various precautions have been taken in the laboratory as well as in the field. In view of your esteemed comment, we have added the following new paragraphs 3 and 4 in section 3.4 to describe our precautions even better:

"*Prior to each day of aerosol particle collection flights, the bellmouth-shaped air inlet, the tube leading to the impactor, the O ring and the two housing halves were cleaned in an ultrasonic bath with soapy water for 15 min, then rinsed with deionized, filtered water (0.3 µm Membrane Filter) and dried in a particle-free environment (laminar air flow box, HEPA H14 filter). Once the parts were dried, the impactor was assembled (excluding sample carrier) and packed together with the inlet into a new, clean, sealable storage bag.*"

"*In the field again, shortly before the particle collection operation, the impactor was taken out of the sealed storage bag, the sample carrier was installed in the impactor and the inlet was plugged onto the tube leading to the impactor. In-between the sampling flights shortly before the next flight operation and shortly before installation of the next unloaded sample carrier, the impactor and the bellmouth-shaped inlet were flushed with filtered air using an battery operated electric blower with a medical ventilation filter installed on its inlet (type Pall Ultipor 100, > 99.999 % retention of airborne bacteria and viruses).*"

Experiments which tested the surface of the bellmouth-shaped inlet with a sticky film have shown that no significant particle numbers were deposited on the inner wall of the inlet and downstream tube, even in case of high particle load in the atmosphere.

**5. In general, the figure legends do not contain enough information for the figure to stand alone without referencing back to the text. Figure 5 is a great example of this.**

We apologize for initially insufficient figure legends and have amended these to provide all necessary information, as can be seen from attachment no. 2.

**6.1 What sort of quantitative data were measured for experiments described in Figure 6 ? There appear to be only qualitative observations. Could you use image-processing tools like IMageJ to formally track the plume of smoke?**

The experiments shown in Figure 6 were conducted to evaluate the best position for the air intake of the particle collection system with regard to isokinetic sampling conditions. As described in section 4.1, the velocity of the air above the copter was calculated by visually tracking characteristic parts of the smoke plume and determining their propagation in consecutive frames (every 40 milliseconds) of the video recording. The results of these experiments are in very good agreement with the computational fluid dynamics (CFD) calculations reported by reference Haas et al. (2014). Nevertheless, as already mentioned in the text at the end of the third paragraph of section 4.1, "…, a more precise determination of the vertical acceleration and velocity of the air flow above the multicopter UAV would be a valuable aspect of future work on this subject.".

**6.2 Did you trap any of the smoke particles on your collection device?**

Answer: No particle collection operation was performed during the smoke experiment. In another experiment performed in the laboratory, the separation ability of the impactor was tested with smoke particles. As a result, a coloured particle deposit was detected on the sample carrier, but not (yet) evaluated. It can be assumed that the smoke particles of the pyrotechnic smoke are very small (approx. 0.1-1 µm). The question whether the newly developed particle collection system is able to separate smoke particles is very interesting, but was not subject of this publication.

**7.1 Your particle trapping efficiency experiments based on two inline trapping surfaces and a single experiment are just not enough. You should aerosolize known particle sizes (such as flourescent microspheres that you can buy at set size ranges), and attempt to trap them on your sampling device. Your efficiency will likely be linked to the size of your particle.**

We really appreciate the encouragement to do more research into the separation efficiency of our impactor.

We are aware that we cannot exclude the possibility that certain particles, whether due to their size and/or due to their morphology, may not be captured by the newly developed particle collection system. And we fully agree that the efficiency depends on the size of the particles. However, as already stated in the title, the present publication is about the collection of "pollen and spores". And in this respect, the results indicate that, in any case the actually collected pollen and spores were trapped with very high efficiency. Originally, we had expected that possibly a

significant part of the impacting pollen and spores would not be retained by the sample carrier of the first impactor, but would detach again, e.g. due to the high mean jet velocity of 50 m s$^{-1}$. Such detached particles, re-entering the air stream, would then have had to be deposited predominantly on the cascaded second impactor, which was not the case.

Two scenarios A and B can be theoretically reflected:
scenario A: The extraction efficiency is 100 %, thus all particles would be extracted in the first impactor, none will reach the second impactor.
scenario B: The extraction efficiency is 50 %, thus 50 % of the particles would be extracted in the first impactor, and 25 % in the second impactor.
The results of the conducted extraction efficiency experiments are much closer to the scenario A than to the scenario B.

In addition to the experiment with the impactor cascade described in this publication, we have also carried out another experiment (not described in the publication) to gain more insight in the extraction efficiency of our impactor:
In this experiment we replaced the downstream impactor with a filter element trapping particles potentially passing by the first impactor stage. These experiments were carried out partly outside in the field, as long as there was enough pollen and spores in the air, and partly in the laboratory using an artificially generated aerosol employing spores of *Lycopodium*. As a result after sampling for 10 minutes, no particles could have been detected on the filter element, whereas the first impact stage was loaded with particles such as pollen grains and spores in the magnitude of about 500 to 1,000 particles.

**7.2 Many of the smaller particles probably go cruising on by the initial trapping surface**

The impactor used in this work is intended to separate particles such as pollen grains and spores with a high separation rate. Other, smaller particles were not of interest within this work.

In general, it can be assumed that impactors optimized for the extraction of pollen grains and spores do not automatically provide the same efficiency for particles that are significantly smaller, e.g. for nanoparticles.

It can be expected that - in order to achieve specified separation efficiency - the smaller the particles, the higher the velocity of the air jet (impaction velocity) must be. The widely used Burkard pollen and spore trap uses a mean impaction velocity of about 5 m s$^{-1}$ to remove pollen from the air stream. For the separation of spores being of a smaller size than pollen, the company recommends a modified inlet that increases the mean flow velocity to about 25 m s$^{-1}$. For comparison: the mean flow velocity in our impactor is 50 m s$^{-1}$.

**7.3 Your final inline sampler could be an impinger, to collect all of the material in a liquid and use that as a basis of quantification**

We are considering your proposal to use an impinger for future experiments but see the problem of reliably detecting a small number of particles in the liquid.

**8.1 Table 1 needs to be overhauled. Order by start time, not by altitude. Also, list stop time of collection.**

Table 1 is intended to give an overview of the measurement flights. For this purpose, the order in the first column by date and in the second column by altitude still seems suitable, especially with regard to the subject of this publication, namely to demonstrate the operational capability of the new system at different flight altitudes.

As it is stated three lines above table 1, that "the particle collection time/duration at each altitude was 10 minutes,…", the "stop time of particle collection" can be easily calculated from the "start time of particle collection" given in table 1. Thus we prefer to indicate only the start time in table 1.

**8.2 Why did the authors choose different sampling times on different days? What is the justification for this? Why not sample the same altitude at multiple times throughout a single day? As it stands, you only present 3 reps of data for 25m and 200 m. 100m is not replicated, and 300 m was only flown twice.**

The measurements presented in this publication primarily serve as a functional demonstration of our new measuring method. Different flight altitudes and sampling times are due to the fact that we wanted to slowly approach the limits of the system. However, all 10-minutes flights started between 2 p.m. and 4 p.m. local time with no rain, temperatures between 15 and 20°C and low wind speed on ground, i.e. comparable atmospheric conditions to be expected (convective conditions). In the future we certainly will focus more on the importance to the comparability of the measurements.

**9. Delete Figure 8. This is really just meant for the discussion.**

According to our opinion, figure 8 provides a very easily accessible overview of the possible influences of the different components of the newly developed particle collection system on the finally determined particle concentration, and thus supports the discussion very vividly.

**10.0 Table 3 needs to be completely overhauled. Consider separate rows for each flight, and separate columns for the pollen and fungi analyzed.**

We have completely revised table 3; please see attachment no. 3.

**10.1 I am concerned about the fungal genera presented in this table. The authors report Puccinia and Epicoccum, but the 'fungal spores' they show in Figure 9 do not appear to be representative of either of these genera.**

Something has been mixed up during manuscript preparation. The reviewer is right. We have amended the following wording (underlined) in second paragraph of section 4.4.1:

"*Only pollen of the genus Taxus, Corylus, Alnus, Cyperaceae, and Salix were counted and listed as well as* *two types of fungal* *spores.* *Fungal spore type 1 probably belongs to the genus Cladosporium, whereas type 2 most likely belongs to the genus* *Epicoccum. Furthermore, charcoal particles with a longitudinal extension of more than 20 µm were also counted. Additionally, a large number of small aerosol particles down to a size of less than 1 µm were visible under the microscope, but are not listed as they cannot be reliably identified by visual inspection only. Figure 9 shows a photograph of the sample carrier content as an example of one of the collection flights.*"

**11. Figure 10 needs to be formatted for publication. I'm not sure what the authors are trying to do here, since they show these data in Table 3.**

Figure 10 has now been revised for publication, as can be seen from the attachment no. 4, and is a visualization of the data of Table 3.

**12. I don't understand the need for Figure 11. Why was 25 m reported? Was this the altitude the reference data were recorded at?**

Figure 11 is a visualization of the comparison of the pollen concentrations of the types Corylus and Alnus measured 1. by us in Poltringen with our newly developed measuring system 25 m a.g.l. and 2. by MeteoSchweiz in Zürich using a Burkard pollen sample near ground level. Unfortunately, no data concerning the concentration of pollen type Taxus and no spore concentration values are provided by MeteoSchweiz, so we had to limit the comparison to the pollen types Corylus and Alnus.

13. Finally, no hypotheses are stated or tested. This makes it very difficult to judge the merits of this work. Did you expect to find different concentrations of pollen at different altitudes? If so, why? How might the concentrations of pollen change throughout a day or night? Did you hover at a single location? What about hovering a multiple locations, but maintaining precise altitudes? More flights are needed to really show the value of this platform. Do you know where the pollen is coming from? Just because a forest is nearby doesn't mean the pollen was coming from there...

The present work describes the development of a new impactor-based particle collection system and its use on a multicopter-based unmanned aerial vehicle. The combination of a high sampling air volume flow of 0.2 $m^3$ per minute and the hovering possibilities of a multicopter UAV provides the potential of particle collection with high temporal and spatial resolution.

And yes, we expected to find different concentrations of pollen at different altitudes, as for example reported by Damialis et. al., 2017, and also by the reference Lin et. al. 2014 that you recommended for citation under item 2. of your comment.

And we also expect a change in pollen concentration between day and night, at least due to the influence of the daily cycle of the atmospheric boundary layer (ABL), e.g. during the morning transition with the dissolution of the nocturnal inversion layer and the formation of a convective layer.

But all these questions were not the subject of the present work, but will be the subjects of future studies working with the newly developed equipment. In any case, the result of the present work provides a valuable tool for investigating all these interesting questions.

**Attachment no. 1**

[revised manuscript text omitted]

**Attachment no. 4**

**figure 10**

[Figure]

[Figure]

[Figure]

---

## Editor Comment (EC1) · Pope (Editor) · 24 Jan 2019

The paper describes a new UAV based particle collection system (PCS) for measurement of bioaerosols in the atmospheric boundary layer. The design and testing of the PCS is rigorous and clearly explained with precise language used throughout. The authors have made great efforts to investigate possible sources of error and bias. The ability and promise of the new system is shown in a series of test flights. The measured bioaerosol concentrations from the test flights compare favourably to collocated measurements. The paper represents a new measurement technique for bioaerosols and should be published after the authors consider the following minor comments.

[Figure]

Abstract and Figure 10 It is unclear what 'charcoal' particulate matter is? Is this black carbon? Black carbon greater than 20 um in size is surprising. Provide more details about what this particulate matter is.

P2 – PM2.5 and PM10 should be defined

P2 – "weighted" should be "weighed"

P2 – the authors may be interested in the work from my group looking at pollen as cloud condensation nuclei, similar to the Hassett et al. 2015 work cited. - Pope, F.D., 2010. Pollen grains are efficient cloud condensation nuclei. Environmental Research Letters, 5(4), p.044015. - Griffiths, P.T., Borlace, J.S., Gallimore, P.J., Kalberer, M., Herzog, M. and Pope, F.D., 2012. Hygroscopic growth and cloud activation of pollen: a laboratory and modelling study. Atmospheric Science Letters, 13(4), pp.289-295.

P3 – It's not clear how the spatial and temporal distribution of pollen would help in paleo reconstruction. Either provide more detail or remove.

P3 L29 – "...to determine how to dimension and where to position..." confusing sentence, reword.

P4 L11 – 200.000 sccm level of precision seems unlikely.

P5 L8 – "...DJI S900 worked reliable and robust" provide more detail. How do you define reliable and robust? How does reliable differ from robust?

P5 L28 – how did you determine that observation of 10 particles was statistically robust?

P6 L1 (and elsewhere) – "isokinetic-near" should be "near-isokinetic".

P7 L11 – define what "lean workflow" means.

P8 L10 – "irrespectively whether" should be "irrespective of whether"

P12 L7 – I'm not sure what "technics" means in this context. Section 3.1 – I found this

section confusing. P12 L10 states that the UAV affects air up to 2 m above it. Figure 6B confirms that air 20 cm above the UAV is definitely affected. Figure 6A shows that air 80 cm above the UAV is somewhat affected. The inlet is positioned 30 cm above the UAV (section 4.1). More rationale is required. Table 1 – Do not use slang "coptor"

P17 L16 – provide more detail to justify the "very good agreement" with the CFD calcualations.

P19 L1 – "no particles would be deposited on the sample..." this is likely true for particles above a certain size. Estimate the size range that the sample procedure is relevant for.

P19 L21 – similar to the previous comment (P19 L1) need to define more precisely the size range the sampling will work for. What do you consider 'small' and 'very small' particles.

P21 L10 – confusing sentence "...as mean values of 25 m, 200 m and 300 m" rewrite.

P21 L1 – the contamination at ground level is going to be dependent on local source, e.g. a pollinating tree nearby could cause significant contamination. Need to provide more nuance. Table 3. Difficult to read the smaller bits of text. Also layout is confusing. Rework to improve readability.] Figure 11 – curved lines are unhelpful since they have no physical meaning, use either straight lines or remove lines completely (my preference).

---

## Author Comment (AC2) · 11 Feb 2019

**Response to comments of Reviewers**

We would like to thank the associated editor Francis Pope for standing in so promptly and serving as the second referee and for his constructive comments which contribute to the quality of our manuscript.

In the following we have addressed all the comments of the Referee. Furthermore, we have amended the manuscript as follows:

Blue: Comments of Reviewer

Black: Answers of Authors

*Black, italic, "": "Changes in the manuscript"*

> Text from manuscript:      *italic*
> Inserted text:             *underlined*
> Removed text:

The amendments made in response to the comments of Referee #1 are not marked-up in the following amendments.

As a supplement, we submit an extract of the manuscript in which all amendments in the text (made in response to both referees as well as minor wording changes and a supplemental Acknowledgement) are marked-up.

**EC1: Francis Pope**

1. Abstract and Figure 10 – It is unclear what 'charcoal' particulate matter is? Is this black carbon? Black carbon greater than 20 μm in size is surprising. Provide more details about what this particulate matter is.

We had deliberately chosen the term "charcoal" to distinguish it from the term "black carbon". And we agree with your comment that black carbon is usually used for small particles with a typical size of less than 1 μm. In contrast, we had used the term charcoal for opaque particles, which appear black in the transmitted light microscope  and which have a wood fibre-like structure and a longitudinal extension of typically more than 20 μm. In order to avoid any misleading association, and in appreciation of your comment, we amended the abstract and the respective

passages as well as Figure 10 and table 3 by now using the term "large opaque particles" instead of "charcoal particles":

page 1, line 19 (abstract):
*"More than thirty aerosol particle collection flights were carried out near Tübingen in March 2017 at altitudes of up to 300 m above ground level (a.g.l.), each with a sampled air volume of 2 m³. Pollen grains and spores of various genera as well as  large (> 20 µm) opaque particles and fine dust particles were collected and specific concentrations of up 20 to 100 particles per m³ were determined by visual microscopic analysis."*

page 23, line 2 (Fig. 9):
*"Only pollen of the genus Taxus, Corylus, Alnus, Cyperaceae, and Salix were counted and listed as well as two types of fungal spores. Fungal spore type 1 probably belongs to the genus Cladosporium, whereas fungal spores type 2 most likely belongs to the genus Epicoccum. Furthermore,  large opaque particles with a longitundinal extension of more than 20 µm were counted; many of these particles having a wood fibre-like structure and the appearance of residues of burned wood or charcoal."*

page 24, line 1 (Fig. 10):
*"The amount of collected pollen grains, fungal spores, and  large (> 20 µm) opaque particles vary significantly between the three sampling days as well as within each sampling day with the respective sampling altitude a.g.l..*

*Only the numbers of the pollen of the genera Taxus, Corylus, and Alnus as well as  large (> 20 µm) opaque particles were high enough (i.e. more than 10 particles per m³) to allow a reliable statistic evaluation."*

**2. P2 – PM2.5 and PM10 should be defined**

Thank you very much for this advice. We have inserted a short definition as follows:

*"The so-called PM2.5 and PM10 particulate matter according to the National Air Quality-Standard for particulate matter of the U.S. Environmental Protection Agency (Vincent, 2007) as well as coarse particles have been chemically characterized by Hueglin et al., 2005. (In simplified view, PM2.5 is the fraction of particulate matter (PM) consisting of inhalable particles having a size of 2.5 µm and smaller, whereas PM10 is the fraction of particulate matter (PM) consisting of inhalable particles having a size of 10 µm and smaller; accordingly, PM2.5 is incorporated in PM10.)"*

**3. P2 – "weighted" should be "weighed"**

Thank you so much for pointing out these misspellings. We have made the necessary corrections.

**4. P2 – the authors may be interested in the work from my group looking at pollen as cloud condensation nuclei, similar to the Hassett et al. 2015 work cited.**

- Pope, F.D., 2010. Pollen grains are efficient cloud condensation nuclei. Environmental Research Letters, 5(4), p.044015. - Griffiths, P.T., Borlace, J.S., Gallimore, P.J., Kalberer, M., Herzog, M. and Pope, F.D., 2012.

- Hygroscopic growth and cloud activation of pollen: a laboratory and modelling study. Atmospheric Science Letters, 13(4), pp.289-295.

Thank you very much for the reference to the further existing research results, by whose quotation the manuscript could be enriched:

*"In meteorology, it is known that mineral dust particles originated from Saharan dust storms and transported for example to Southern Florida effectively act as ice nuclei being capable for glaciating super cooled altocumulus clouds (Sassen et al., 2003). Pollen grains, although being only moderately hygroscopic, are able to act as cloud condensation nuclei and exhibiting a bulk uptake of water under subsaturated conditions (Pope, 2010). Investigations on the hygroscopic growth of pollen suggest that extreme pollen concentrations (> 1,000 m$^{-3}$) may interfere with the activation of the background sulphate aerosol mode in pristine environments (Griffiths et al., 2012). Also spores of which millions of tons are dispersed into the atmosphere every year, may act as nuclei for condensation of water in clouds (Hassett et al., 2015)."*

**5. P3 – It's not clear how the spatial and temporal distribution of pollen would help in paleo reconstruction. Either provide more detail or remove.**

Thank you very much for your advice, which has enabled us to further improve clarity of the manuscript. We have amended the text as follows:

*"Here, for example, the knowledge of the spatial and temporal distribution of pollen could help to gain insights in their genus-specific propagation behaviour and possible transport distances. This would enable to improve the accuracy of paleoclimate models derived from pollen grains retrieved from extracted from lacustrine or marine sediments (Shang et al., 2009)."*

**6. P3 L29 – ". . .to determine how to dimension and where to position. . ." confusing sentence, reword.**

Thank you very much for your advice, which has enabled us to further improve clarity of the manuscript. We have amended the text as follows:

*"The experimental results were used to determine  the dimension and  position of the air intake of the PCS on the multicopter UAV in order to provide substantially isokinetic sampling conditions."*

**7. P4 L11 – 200.000 sccm level of precision seems unlikely.**

Sorry, our mistake. We used the wrong decimal separator. To avoid any further misunderstandings we amended the text as follows:

*"(corresponding to 200 thousand standard cubic centimetres per minute – 200,000 sccm )"*

**8. P5 L8 – "...DJI S900 worked reliable and robust" provide more detail. How do you define reliable and robust? How does reliable differ from robust?**

We have deliberately used the terms reliable and robust to describe the following observations: The DJI S900 worked reliably, i.e. not a single flight interruption due to technical problems occurred. And it was robust, i.e. the components withstood all applied stresses, for example during harder landings, without any problems or hardware failure; as a result, all flights described in this manuscript – and even more – have been performed with the very same multicopter UAV model.

In appreciation of your comment, we amended the last sentence of the first paragraph of section 2.1 as follows:

*"At ambient air temperatures between -5 °C and +37 °C as experienced during tens of flight operations in 2017, the DJI S900 worked  reliably, i.e. not a single flight interruption due to technical problems occurred, and it was robust, i.e. the components withstood all applied stresses without any problems or hardware failure.*

**9. P5 L28 – how did you determine that observation of 10 particles was statistically robust?**

We apologise that the term "statistically evaluable" is not clear enough.
Our considerations in this context are that – although the utmost care has been taken in the preparation and examination of the sample carriers – an incorrect occurrence of a single hit of a particle, or of a single loss of a particle, cannot be completely ruled

out. In order to keep the corresponding error below 10 %, the aim within this study is to collect at least 10 particles of each genus. To ensure this even in the case of a particle concentration in the sampled air being as low as 5 particles per m$^3$, an air volume of 2 m$^3$ has to be sampled. Correspondingly, we did not considered particles that were counted less than 10 times.

Nevertheless, to avoid any lack of clarity, we amended the text as follows:

*"A new PCS was developed in order to meet the requirements for aerial use onboard a multicopter UAV. To ensure a  number of at least 10 collected particles even in the case of a particle concentration in the sampled air being as low as 5 particles per m$^3$, an air volume of 2 m$^3$ has to be sampled."*

**10. P6 L1 (and elsewhere) – "isokinetic-near" should be "near-isokinetic".**

Thank you so much for pointing out these misspellings. We have made the necessary corrections.

**11. P7 L11 – define what "lean workflow" means.**

Thank you for pointing that out. By a lean workflow we mean a process that is optimized in terms of complexity, equipment requirements, and time expenditure. Working steps should be kept as simple as possible in order to avoid errors and thus optimize the applicability of the method. In order to enhance the clarity of our considerations we amended the text as follows:

*"Additionally, in order to achieve a lean workflow from sampling to visual particle identification and counting, the extracted particles should be easily accessible for visual analysis without complex and time-consuming sample preparation steps. In this context "lean workflow" also means that preferably an initial estimate of the quantity and type of particles collected should be possible already in the field by visual inspection with simple tools such as a magnifying glass; this allows, if necessary, an adjustment of the flight altitude or the sampling operation period during the immediately following particle collection flight."*

**12. P8 L10 – "irrespectively whether" should be "irrespective of whether"**

Thank you so much for pointing out these misspellings. We have made the necessary corrections.

13. P12 L7 – I'm not sure what "technics" means in this context. Section 3.1 – I found this section confusing. P12 L10 states that the UAV affects air up to 2 m above it. Figure 6B confirms that air 20 cm above the UAV is definitely affected. Figure 6A shows that air 80 cm above the UAV is somewhat affected. The inlet is positioned 30 cm above the UAV (section 4.1). More rationale is required.

Thank you for bringing this to our attention. We apologize for the lack of clarity. Section 3.1 is now revised to make it more clear that the Figures 6A and 6B are intended to show that – in accordance with the CFD calculations reported by Haas et al. (2014) – the air volume mixed by the propellers of the multicopter UAV extends only about 2 m above the multicopter UAV: the smoke plume approaching 1.8 m above the multicopter UAV (Fig. 6B, upper smoke plume) is already only very slightly affected, and the smoke plume approaching 2.4 m above the multicopter UAV (Fig. 6A, middle smoke plume) remain unaffected.

Section 3.1:

[revised manuscript text omitted]

**15. Table 1 – Do not use slang "coptor"**

Many thanks for this indication on the inappropriately shortened spelling; we have replaced the term "copter" with "multicopter" in Table 1.

**16. P17 L16 – provide more detail to justify the "very good agreement" with the CFD calculations.**

As already mentioned in the response to the comments to section 3.1, we have amended section 4.1 as follows:

Section 4.1: first paragraph
*The air flow pattern or s"Smoke pPlume tTests" (Sect. 3.1) carried out allow a quantitative determination of the air flow velocities,. however, with underline{Despite their} limited resolution, only. Nevertheless, the results obtained underline{here} are in very good agreement with the CFD calculations reported by Haas et al. (2014).: underline{The smoke plume approaching 20 cm above the propellers of}*

*the multicopter UAV is directly captured by the propellers (Fig. 6B, middle smoke plume). Also the smoke plume approaching 80 cm above the multicopter UAV is strongly affected and accelerated downwards (Fig. 6A, lower smoke plume). The smoke plume approaching 1.8 m above the multicopter UAV, on the other hand, is already only very slightly affected (Fig. 6B, upper smoke plume). And the smoke plume approaching 2.4 m above the multicopter UAV remains unaffected (Fig. 6A, middle smoke plume). Thus, these results correspond very well with the CFD-calculations reported by Haas et al. (2014), according to which the air volume mixed by the propellers of the multicopter UAV extends only about 2 m above the multicopter UAV. In addition, Fig. 6B also shows that the air volume mixed by the propellers extends further below the multicopter UAV than above the multicopter UAV, as predicted by the CFD-calculations.”*

17. P19 L1 – “no particles would be deposited on the sample...” this is likely true for particles above a certain size. Estimate the size range that the sample procedure is relevant for.

We agree totally with your comment, but on page 19, line 1, the “no particles” statement refers to a theoretical model with “an ideal extractions efficiency of 100 %” introduced in the previous line 35 on page 18. Therefore we kept the wording at this point, but amended the following text (see next comment) according to your advice.

18. P19 L21 – similar to the previous comment (P19 L1) need to define more precisely the size range the sampling will work for. What do you consider ‘small’ and ‘very small’ particles.

We have amended the manuscript as follows:

*“The particle extraction and retention capability of the newly developed PCS was demonstrated for pollen of the genera Taxus, Alnus, and – with restrictions concerning statistical data base –  Corylus and Pinus, which were present in the air at the time of the extraction efficiency experiment. While the number of pollen grains of the  genera Corylus and Pinus  are regarded of being too small for a statistical evaluation, the number of pollen grains of the genus Taxus and Alnus collected in upstream impactor no. 1  were about 100 to 250 times the number of corresponding particles in downstream impactor no. 2. As a result, the extraction efficiency, or retention ratio, of the impactor under the given conditions (200 litres per minute) concerning the pollen grains of genera Taxus, Alnus, and Corylus is at least 98 %.*

*With regard to the question whether this high extraction and retention rate also applies to other particles, it should be noted that in the widely used Burkard pollen trap a mean jet velocity of 6 m s$^{-1}$ is sufficient enough to reliably extract pollen grains and spores from the air. For the widely used Burkard pollen traps, a modified orifice with a reduced width of 0.5 mm is available, which increases the mean jet velocity to 24 m s$^{-1}$ in order to improve the trapping efficiency for particles in the range 1-10 µm diameter (Datasheet Burkard 7 Day Recording Volumetric Spore Sampler, Burkard Scientific). As  shown in Figure 9, the newly developed impactor (working with a mean jet velocity of 50 m s$^{-1}$)  extracts aerosol particles having a size between the resolution limit of the light microscope (being in the range of 1 µm) and approximately 60 µm*
*Further investigations are necessary to check whether the high extraction rates (of at least 98 %) determined for pollen of the genera Taxus, Alnus, and Corylus (with a typical size between 20 and 30 µm) also apply to particles in the µm and sub µm range."*

19. P21 L10 – confusing sentence ". . .as mean values of 25 m, 200 m and 300 m" rewrite.

20. P21 L1 – the contamination at ground level is going to be dependent on local source, e.g. a pollinating tree nearby could cause significant contamination. Need to provide more nuance.

Thank you very much for this note. We have added additional information and clarified the confusing sentence as follows:

*"Particle contamination is a potential error source that leads to higher particle numbers deposited on the sample carrier. Within the present study, experiments concerning potential contamination on ground as well as particle contamination during ascent and descent of the multicopter UAV were performed. Concerning potential particle contamination on ground, in total 4 pollen grains were identified on the sample carrier, i.e. 2 of the genus Taxus, 1 of the genus Alnus, and 1 of the genus Corylus, as the result of a 15 minutes exposure of the uncovered sampling carrier to the ambient air. This small number is certainly also due to the lack of local sources such as pollinating trees or bushes within a radius of 150 m around the location of exposure (airfield in Poltringen).*

*For the evaluation of these results, the concentration of the pollen grains in the ambient air must be taken into account. The contamination experiments were carried out on March 10, 2017 at the same time as the aerosol particle collection flights.  The mean values of the concentrations measured at the three altitudes (25 m, 200 m, and 300 m a.g.l.) are: 53 pollen grains per m$^3$ of the genus Taxus, 44 pollen grains per m$^3$ of the genus Alnus, and 16 pollen grains per m$^3$ of the genus Corylus  (Tab.le 3)."*

**Table 3. Difficult to read the smaller bits of text. Also layout is confusing. Rework to improve readability.**

Thank you very much for this advice. We do assume that your review refers to the originally submitted manuscript, but have already revised the diagram in response to the review of referee #1 as shown below. If a further revision is necessary, we politely ask for a corresponding brief notice; we will then do this further revision at short notice.

**Results of the Measurements performed at Poltringen Airfield with the newly developed Particle Collection System mounted on the multicopter UAV**

|  | March 3 | | | | March 10 | | | March 16 | | |
|---|---|---|---|---|---|---|---|---|---|---|
| collection start time (local time) | 15:55 | 15:20 | 15:05 | 15:44 | 14:57 | 14:38 | 15:40 | 14:18 | 13:55 | 14:40 |
| flight altitude (in m a.g.l) | ground | 25 m | 100 m | 200 m | 25 m | 200 m | 300 m | 25 m | 200 m | 300 m |
| *Taxus* | 32 | 22 | 24 | 2 | 113 | 133 | 70 | 135 | 175 | 88 |
| *Corylus* | 27 | 35 | 30 | 29 | 32 | 36 | 26 | 4 | 1 | - |
| *Alnus* | 128 | 167 | 159 | 181 | 109 | 91 | 63 | 18 | 11 | 12 |
| *Cyperaceae* | - | - | - | - | 5 | 2 | 2 | - | - | - |
| *Salix* | - | 3 | 2 | 1 | 9 | 3 | 5 | 23 | 3 | 10 |
| fungal spores type 1 | 22 | 5 | 17 | 2 | 200 | 114 | 131 | 2 | 2 | 2 |
| fungal spores type 2 | 16 | 1 | 3 | 4 | 3 | 4 | 4 | 2 | 5 | 3 |
| opaque particles >20 µm | 2 | 11 | 4 | 9 | 52 | 33 | 26 | 30 | 34 | 16 |

**Comparison to the Statement of the Deutscher Polleninnformationsdienst (PID)**

| | Statement of the Deutscher Polleninformationsdienst (PID) "Wochenpollenvorhersage" | | |
|---|---|---|---|
| | Week of March 1, 2017 (KW9) | Week of March 8, 2017 (KW10) | Week of March 15 (KW 11) |
| Pollen of the genera *Taxus* | first weak load "erste schwache Belastung" | short time large amount "kurze Zeit große Menge" | the most abundant genus of Pollen "die mengenmaäßig haäufigste Pollenart" |
| Pollen of the genera *Corylus* and *Alnus* | first high concentration "erstmals hohe Konzentration" | approaches the end "nähert sich dem Ende" | fadet ("abgeblüht") |

**Comparison to the Measurements of MeteoSchweiz performed in Zürich**

| | March 3, 3017 | March 10, 2017 | March 16, 2017 |
|---|---|---|---|
| *Alnus* | Number of pollen grains per m$^3$ | | |
| PCS in Poltringen (25 m a.g.l.) | 18 | 15 | 2 |
| Burkard sampler in Zürich | 41 | 20 | 5 |
| *Corylus* | Number of pollen grains per m$^3$ | | |
| PCS in Poltringen (25 m a.g.l.) | 84 | 55 | 9 |
| Burkard sampler in Zürich | 39 | 45 | 8 |

Figure 11 – curved lines are unhelpful since they have no physical meaning, use either straight lines or remove lines completely (my preference).

Thank you very much for this advice. We have revised the diagram accordingly. To further improve readability, in both diagrams blue markings are now used for the values measured in Zurich, red markings for the values measured in Poltringen.

[revised manuscript text omitted]